# Multipole groups and fracton phenomena on arbitrary crystalline lattices

Daniel Bulmash, Oliver Hart ⬤, and Rahul Nandkishore
*Department of Physics and Center for Theory of Quantum Matter,*
*University of Colorado Boulder, Boulder, Colorado 80309 USA*
(Dated: January 25, 2023)

Multipole symmetries are of interest in multiple contexts, from the study of fracton phases, to nonergodic quantum dynamics, to the exploration of new hydrodynamic universality classes. However, prior explorations have focused on continuum systems or hypercubic lattices. In this work, we systematically explore multipole symmetries on arbitrary crystal lattices. We explain how, given a crystal structure (specified by a space group and the occupied Wyckoff positions), one may systematically construct all consistent multipole groups. We focus on two-dimensional crystal structures for simplicity, although our methods are general and extend straightforwardly to three dimensions. We classify the possible multipole groups on all two-dimensional Bravais lattices, and on the kagome and breathing kagome crystal structures to illustrate the procedure on general crystal lattices. Using Wyckoff positions, we provide an in-principle classification of all possible multipole groups in any space group. We explain how, given a valid multipole group, one may construct an effective Hamiltonian and a low-energy field theory. We then explore the physical consequences, beginning by generalizing certain results originally obtained on hypercubic lattices to arbitrary crystal structures. Next, we identify two seemingly novel phenomena, including an emergent, robust subsystem symmetry on the triangular lattice, and an exact multipolar symmetry on the breathing kagome lattice that does *not* include conservation of charge (monopole), but instead conserves a vector charge. This makes clear that there is new physics to be found by exploring the consequences of multipolar symmetries on arbitrary lattices, and this work provides the map for the exploration thereof, as well as guiding the search for emergent multipolar symmetries and the attendant exotic phenomena in real materials based on nonhypercubic lattices.

## CONTENTS

## I. INTRODUCTION

Multipole symmetries have become a major topic of interest in condensed matter physics, quantum dynamics, and quantum information. This began with the work of Pretko [1] identifying conserved multipole moments as underlying the

exotic phenomenology of fractons (for reviews, see Refs. [2–4]). Subsequently, new field theories with fractonic excitations and general multipolar symmetry groups were written down and systematized [5, 6]. The new thermodynamic phases enabled by such symmetries and their spontaneous breaking remain topics of active exploration [7–10]. In a parallel development, it was realized in Ref. [11] that imposing multipole symmetries on quantum dynamics could give rise to ergodicity breaking. This was later explained in terms of Hilbert space shattering/fragmentation [12–14], a phenomenon that has been observed experimentally [15, 16], which could be harnessed for both quantum memories [12] and metrology [17], and which has been undergoing intensive exploration [18–31]. In a third development, it was realized in Refs. [32, 33] that multipolar symmetries could lead to new hydrodynamic universality classes. This has initiated yet another line of research exploring novel hydrodynamics with multipolar symmetries [34–43]. Common to all these distinct research programs is the central role of multipole symmetries.

Prior explorations of multipole symmetries have largely been limited to either systems in the continuum or on hypercubic lattices. It is worth noting that formulating the problem in the continuum introduces certain pathologies – for instance, the dipoles and multipoles are not quantized and there appear a continuous infinity of particle types and superselection sectors. How to properly define the problem in the continuum is a program of ongoing research [44]. In practice, most works implicitly assume an underlying lattice. When lattice symmetries are treated seriously, the problem is almost always formulated on a square or cubic lattice, with rare exceptions (e.g., [45]). However, given the wide range of crystal structures, it is natural to wonder what happens if we formulate the problem on an arbitrary crystal structure that is not a hypercubic lattice. Multipole symmetries are *not* purely internal, and mix with spatial transformations [6], so the extension to arbitrary lattices is decidedly nontrivial. As a simple example to illustrate this nontriviality, let us work in one dimension and imagine imposing dipole conservation, but not monopole conservation. This automatically implies that the theory must lack translation invariance – the dipole moment is defined with respect to an origin, and monopole charge may be freely added or removed at that origin without changing the dipole moment. Conversely, if we wish to retain translation symmetry, and conserve dipole moment, then we must also conserve monopole moment (charge). What happens if we move up from one dimension to higher dimensional lattices? How does the set of multipole symmetries that can be consistently imposed depend on the choice of lattice? Are there qualitatively new phenomena that arise once we move away from continuum systems or hypercubic lattices? These questions remain largely open, and could provide routes to the design of new fracton phases on non-hypercubic lattices, to the realization of new kinds of nonergodic dynamics, and to the identification of new hydrodynamic universality classes. They would also guide our search for such phenomena in real materials – since while multipolar symmetries may emerge in the low-energy description of a real material, *which* multipolar symmetries emerge would depend on the crystal structure of the material, which may not be hypercubic.

In this work, we undertake a systematic exploration of multipole symmetries on arbitrary lattices. We explain how, given a space group symmetry and a set of occupied Wyckoff positions (which together determine the crystal structure), one may systematically construct all possible consistent multipole symmetry groups to any desired order. While we work in two dimensions for simplicity, the methods we develop are general and should extend to three (or higher) dimensions *mutatis mutandis*. For all two-dimensional Bravais lattices, we exhaustively classify all possible consistent multipole groups at order $n = 1$ (dipole), $n = 2$ (quadrupole), $n = 3$ (octupole) and $n = 4$. We explain how the classification may be extended beyond Bravais lattices to deal with bases and nonsymmorphic symmetries, thereby allowing us to access arbitrary wallpaper groups. We also explain that the space group itself is not sufficient to fully specify the problem – one needs the full crystal structure, which also involves knowledge of the occupied Wyckoff positions. We illustrate the general framework by an explicit computation of the consistent multipole groups up to order $n = 2$ on the kagome and breathing kagome crystal structures. This gives an in-principle classification of all possible multipole groups in any space group.

While the first part of this manuscript is essentially mathematical in nature, classifying the possible consistent multipole groups on various crystal structures, the second part of this manuscript discusses the physical consequences. We begin by discussing how knowledge of the multipole group may be used to write down effective low-energy field theories, and how these may be discretized to yield effective Hamiltonians on the lattice in question. We then discuss how this framework may be used to generalize certain results originally obtained on hypercubic lattices. For instance, we present a general understanding of a minimal set of symmetries that must be imposed to yield localization on arbitrary crystal lattices, generalizing the results of Ref. [29]. Then we discuss two seemingly novel phenomena that arise when we move beyond hypercubic lattices (a) an emergent robust subsystem symmetry arising on the triangular lattice, and (b) an unusual situation arising on breathing kagome, where one can obtain multipolar conservation laws without conservation of monopole charge, but while retaining translation symmetry (and also an emergent vector conserved charge). These two examples are not exhaustive, but illustrate that new physics can arise when one generalizes away from hypercubic lattices. The systematic exploration of new physics arising from multipole symmetries on arbitrary crystal structures therefore promises to be a fertile territory for exploration, and this manuscript provides the map.

This manuscript is structured as follows. We start by introducing the methodology for deriving multipole groups that are compatible with the space group of the lattice in Sec. II. For readers not interested in the technical details, we begin this section by presenting an intuitive overview of the general procedure. We then apply the formalism to the five Bravais lattices in two dimensions in Sec. II B, and to generic wallpaper groups in Sec. II C, where a number of additional complexities arise. The second half of the manuscript is concerned with exploring the consequences of the multipole groups found in Sec. II. First, we describe how to construct local Hamiltonians

that are invariant under space group operations and conserve multipole moments belonging to a particular multipole group in Sec. III. Then, in Sec. IV, we extend some classic results derived on hypercubic lattices to arbitrary crystal structures, and also present two examples of interesting phenomena that can arise when one goes beyond simple hypercubic lattices. We close with a discussion of our results in Sec. V. Finally, some clarification on notation. Throughout the manuscript, we use the physics convention for the dihedral group: $D_M$ is the group of symmetries of a regular $M$-gon. Similarly, our notation for the irreducible representations (irreps) of $D_M$ is set out in Appendix A.

## II. FORMALISM FOR DERIVING LATTICE MULTIPOLE GROUPS

The goal of this section is to build a procedure that algorithmically determines the constraints that space group symmetries place on lattice multipole groups. More precisely, suppose the multipole group contains a particular polynomial shift symmetry $f$. Then we wish to determine: what additional polynomial shift symmetries must appear in the multipole group in order to preserve the space group symmetry?

We will first describe the general procedure. Then, we will implement it for all five Bravais lattices in 2D, explaining several details and subtleties that arise in the different examples. Finally, we will explain through examples several important concepts that appear when classifying multipole groups on a crystalline lattice with a basis. Using Wyckoff positions, we will give an in-principle classification of all possible multipole groups in any space group.

### A. General procedure

The input to our procedure is a space group $S$, its action on a given basis of the crystalline lattice $\mathcal{L}$, and a nonnegative integer $n$. The output is a list of all multipole groups $\mathcal{M}$ compatible with the space group $S$ (in the sense the quotient of $\mathcal{M}$ by its subgroup of pure polynomial shift symmetries is $S$) that contain polynomial shift symmetries of degree at most $n$.

We place a scalar field $\phi_i(\mathbf{r})$ at each basis location in the crystalline lattice. Here $i$ labels a basis element of $\mathcal{L}$, and, depending on the physics, $\mathbf{r}$ may either be the position of a lattice site or a basis site; we will discuss the distinction in Sec. II C 3. In what follows, we assume for simplicity that, given an element $\mathbf{s} \in S$, the field transforms as

$$\phi_i(\mathbf{r}) \xrightarrow{\mathbf{s}} \phi_{\mathbf{s}(i)}(\mathbf{s}(\mathbf{r})), \qquad (1)$$

where $\mathbf{s}(i)$ is some permutation of the lattice basis elements and $\mathbf{s}(\mathbf{r})$ is some action on the spatial coordinates. More generally, the field $\phi$ need not transform as a scalar under space group operations. This more general case is easily incorporated into the formalism; we give an example in which the field transforms as vector and discuss some consequences in Appendix B. Under

an infinitesimal polynomial shift symmetry $f$, we have

$$\phi_i(\mathbf{r}) \xrightarrow{f} \phi_i(\mathbf{r}) + \lambda f_i(\mathbf{r}), \qquad (2)$$

where the $f_i(\mathbf{r})$ are a collection of polynomials that we assume to have degree at most $n$, and $\lambda$ is the symmetry parameter. A symmetry under such a polynomial shift implies conservation of the corresponding multipole moment. We demand that the set of polynomial shift symmetries be closed under the action of the space group – a defining property of the multipole group [6]. Commuting $\mathbf{s}$ through the polynomial shift symmetry, we see that if $f$ is in $\mathcal{M}$, then $\mathcal{M}$ must also contain

$$\phi_i(\mathbf{r}) \to \phi_i(\mathbf{r}) + \lambda \left( f_{\mathbf{s}(i)}(\mathbf{s}(\mathbf{r})) - f_i(\mathbf{r}) \right). \qquad (3)$$

Given $f$, then, we must find its orbit under the space group. While this problem is well-posed, the space group has infinite order, although it is finitely generated (e.g., there are an infinite number of symmetry translations, but they are all generated by a finite set of basis vectors), and the classification problem becomes awkward[1]. However, in general, space group elements act on spatial coordinates as

$$\mathbf{s}(\mathbf{r}) = M_\mathbf{s}\mathbf{r} + \mathbf{t_s}, \qquad (4)$$

where $M_\mathbf{s}$ is a matrix acting on the components of $\mathbf{r}$ and $\mathbf{t_s}$ is some constant vector. Therefore, if we replace $\mathbf{s}(\mathbf{r})$ with a modified transformation $\tilde{\mathbf{s}}$ that acts as

$$\tilde{\mathbf{s}}(\mathbf{r}) = M_\mathbf{s}\mathbf{r}, \qquad (5)$$

that is, we completely drop the translation, then applying this transformation to *homogeneous* polynomial of degree $k$ produces another *homogeneous* polynomial of degree $k$. We can therefore sort the set of homogeneous polynomials $f_i(\mathbf{r})$ into irreps of the *finite* group generated by the transformations

$$f_i(\mathbf{r}) \xrightarrow{\tilde{\mathbf{s}}} f_{\mathbf{s}(i)}(\tilde{\mathbf{s}}(\mathbf{r})), \qquad (6)$$

which is a straightforward task that can be done algorithmically. There are several ways to accomplish this using standard group-theoretic methods; see Appendix A for a numerical method using character theory and an analytical method using Clebsch-Gordon coefficients for discrete groups.

We call the group of modified transformations the "extended point group"; the reason for the terminology is that this group evidently contains the point group but, for a non-symmorphic symmetry group, is in general larger than the point group. The extended point group is in fact isomorphic to the quotient $S/T$, where $T \subset S$ is the set of translations, but the extended point group is implemented slightly differently because it discards

---

[1] The basic technical problem is the following. Consider the vector space spanned by monomials, equipped with the natural inner product where different monomials are orthogonal. Then translations do not act as orthogonal operators on that vector space. We would thus be forced to consider non-orthogonal representations of the space group, which is a more challenging problem than the representation theory considered in this paper.

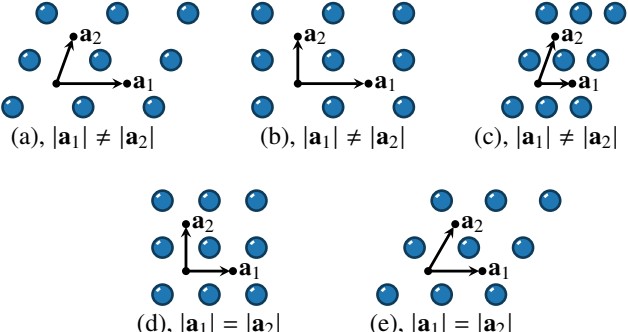

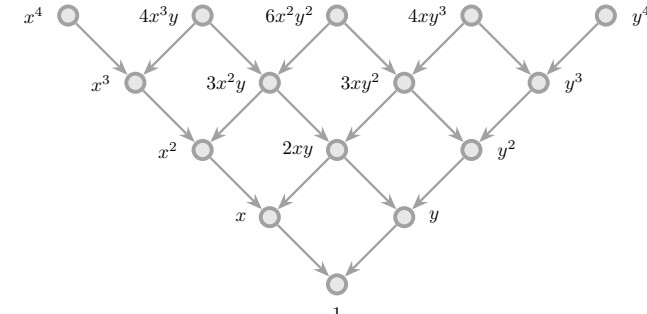

FIG. 1. Illustration of the five Bravais lattices in two spatial dimensions. The lattice translation vectors $\mathbf{a}_1$ and $\mathbf{a}_2$ are denoted by black arrows, which make an angle $\gamma$ with one another. (a) Monoclinic, with point group $C_2$, (b), (c) orthorhombic, with point group $D_2$, (d) square, with point group $D_4$, and (e) triangular ($\gamma = \pi/3$), with point group $D_6$.

FIG. 2. Mixing of monomials under lattice translations for the monoclinic and orthorhombic Bravais lattice types. Every arrow means that some (infinitesimal) translation acting on the monomial of higher degree generates the monomial(s) of lower degree. That is, if a particular monomial is included in the multipole group, compatibility with space group operations requires the multipole group to contain all lower-order monomials that can be reached by following an arrow from the original monomial. Linear combinations of degenerate irreps must be dealt with separately, as described in the main text.

both integer and fractional translations.

This does not yet solve the problem, because under a true space group operation (including translations), a representation of degree $k$ will generically produce terms of all lower degrees. However, the terms of degree $k$ will stay within the representation we found earlier because translations only produce terms of degree less than $k$. Hence, under a true space group operation, the representations of degree $k$ "mix" with representations of degree less than $k$, but do not mix with other representations of degree $k$. In fact, we show in Appendix C that for each representation of degree $k$, we may take an *infinitesimal* translation by $\mathbf{t}_s$, which produces homogeneous polynomials of degree $k-1$ only. By iterating this procedure on all representations of degree less than $k$, we find the full set of polynomials that appear under these translation operations.

To summarize, we sort homogeneous polynomials into irreps of the extended point group (6), and then see how they "mix" under the translation piece of each space group element. Under (true) space group transformations, an element $f$ of a fixed irrep produces other elements of its irrep and linear combinations of its "descendant" irreps. If $f \in \mathcal{M}$, then the entire irrep to which $f$ belongs, and all of the descendants of that irrep, must appear in $\mathcal{M}$ as well.

This procedure can be generalized to the case where $f$ is an arbitrary linear combination of elements from different irreps of the extended point group. We will see in the context of various examples that this generalization can be subtle.

Before proceeding to examples, we comment that, for convenience, we have assumed that the multipole group contains polynomial shift symmetry for scalar fields, but the approach would work equally well for, say, spins, e.g., rotating

$$\hat{\mathbf{S}}(\mathbf{r}) \rightarrow e^{i\hat{S}_z(\mathbf{r})f(\mathbf{r})}\hat{\mathbf{S}}(\mathbf{r})e^{-i\hat{S}_z(\mathbf{r})f(\mathbf{r})} \tag{7}$$

where $f(\mathbf{r})$ is some polynomial. We will elaborate further on this context in Sec. III A.

### B. Bravais lattices

When working with Bravais lattices, there is no basis index $i$ in, e.g., Eq. (1). As a result, the procedure outlined above becomes straightforward. Extended point group operations (6) are exactly point group operations, and so we just need to sort homogeneous polynomials into irreps of the point group $P$. The "mixing" of irreps then comes from translations by lattice vectors. We will find the possible multipole groups compatible with all five Bravais lattices in two dimensions (depicted in Fig. 1), starting with the lattice that possesses the fewest symmetries. As we increase the size of the symmetry group, various additional complexities will appear; while the formalism will remain unchanged, we will highlight some subtleties in its implementation.

#### 1. Monoclinic

We begin with the monoclinic Bravais lattice [Fig. 1(a)], whose point group is $C_2$, with generator $(x, y) \leftrightarrow (-x, -y)$. In crystallographic notation, which will be used throughout the paper, the space group is $p2$. By inspection, each monomial $x^m y^n$ (with $m, n \geq 0$) of even degree forms a trivial irrep of $C_2$ and each monomial of odd degree forms the nontrivial one-dimensional irrep of $C_2$.

After sorting polynomials into irreps of the point group, we must determine what constraints translations put on the multipole group. Suppose that we include $x^m y^n$ in the multipole group for a given pair $m$, $n$. As shown in Appendix C, we must also include shift symmetries obtained by infinitesimal translations of $x^m y^n$ along the lattice directions. Since there are two linearly independent translations and polynomial shift symmetries can have any real (not just integer) coefficients, it suffices to choose the translations to be along $x$ and $y$ (denoted

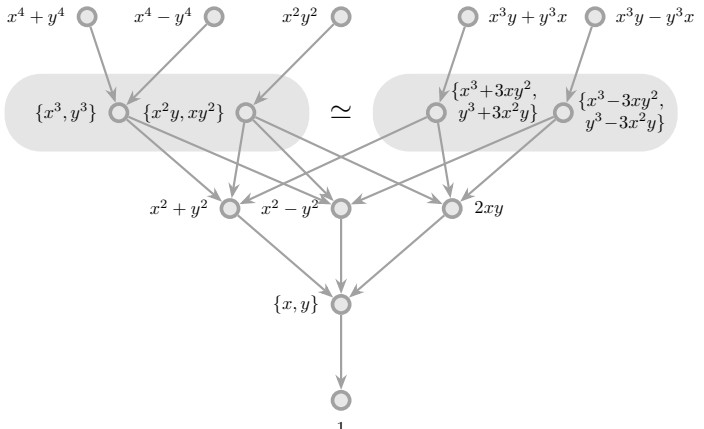

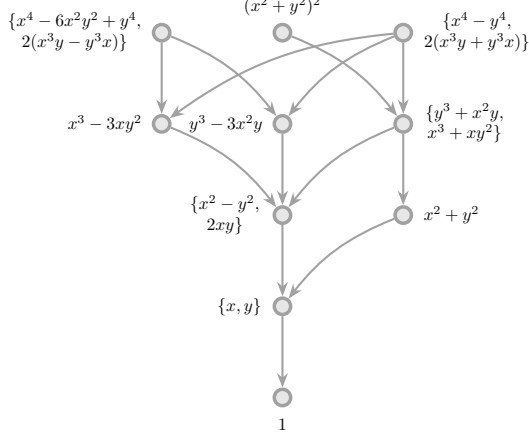

FIG. 3. Mixing of irreps of the point group $D_4$ by translation symmetry for the square lattice. Every arrow means that some (infinitesimal) translation acting on the polynomial(s) of higher degree generates the polynomial(s) of lower degree. All multipole groups that contain a polynomial of degree $n \geq 2$ must also include both components of dipole, and charge. The two gray regions at order three indicate that there are two alternative bases for cubic polynomial shift symmetries. The representations of the polynomials under the point group are given in Table I.

FIG. 4. Mixing of irreps of the point group $D_6$ by translation symmetry for the triangular lattice. Every arrow means that some (infinitesimal) translation acting on the polynomial(s) of higher degree generates the polynomial(s) of lower degree. As for the square lattice, all multipole groups that contain a polynomial of degree $n \geq 2$ must also include both components of dipole, and charge. The representations of the polynomials under the point group are given in Table II.

$T_x$ and $T_y$, respectively), even if the discrete lattice translations are not orthogonal. Hence, including $x^m y^n$ requires us to include $x^{m-1} y^n$ (if $m > 0$) and $x^m y^{n-1}$ (if $n > 0$) in the multipole group.

Iterating this procedure, we find that including $x^{m-1} y^n$ in the multipole group requires $x^{m-2} y^n$ (if $m > 1$) and $x^{m-1} y^{n-1}$ (if $n > 0$) to appear in the multipole group as well. Each irrep of degree $k$ therefore has a set of "descendants" of degree $k - 1$, and the descendants have their own descendants, and so on. We can organize this information into a graphical tree-like structure, presented in Fig. 2. The meaning is that if we include any one monomial in $\mathcal{M}$, then all of its descendants (i.e., any monomial that can be reached by following a series of arrows starting at the original monomial) must also appear in $\mathcal{M}$.

We now address an important subtlety. Each irrep of the point group is highly degenerate, in the sense that any linear combination of even- (or odd-) degree monomials also forms an irrep of $C_2$. This is unimportant if we include in $\mathcal{M}$ a linear combination of polynomials transforming under different irreps; in that case, we have to include both polynomials or neither in the multipole group. For example, including $x^4 + 3xy^2$ as a generator in $\mathcal{M}$ also forces us to include $x^4 - 3xy^2$ since one term is odd under rotations and one term is even. The polynomials $\{x^4 + 3xy^2, x^4 - 3xy^2\}$ span the same set of polynomials as $\{x^4, 3xy^2\}$, so we can just include both monomials. To ensure the multipole group is closed under translations, we should then include the descendants of both monomials.

However, if we include a linear combination of two identical representations, we may only need to include corresponding linear combinations of the descendants. For example, $x^3 y + xy^3$ only forces us to include the span of $\{3x^2 y + y^3, x^3 + 3xy^2\}$ (and lower-degree descendants), not the span of $\{x^3, y^3, 3x^2 y, 3xy^2\}$.

The tree in Fig. 2 is therefore still useful for finding the possible multipole groups, but one must be careful when including linear combinations of polynomials transforming under the same irrep of the (extended) point group.

### 2. Orthorhombic

Next, we consider Bravais lattices of orthorhombic type, for which the point group is $D_2$ [Figs. 1(b), (c)]. For simplicity we choose coordinates so that the point group is generated by reflections about the $y$- and $x$-axes. Again, each monomial forms an irrep of the point group. Choosing the generating mirror of $D_2$ to send $y \leftrightarrow -y$, the monomial $x^n y^m$ forms the 1D irrep $A_{(-1)^{n+m},(-1)^m}$ where the notation is set in Appendix A; to briefly summarize, the first index is the eigenvalue under two-fold rotations [i.e., $(x, y) \leftrightarrow (-x, -y)$] and the second index is the eigenvalue under the generating mirror.

For the rectangular Bravais lattice (space group $pmm$), the monomial $x^m y^n$ transforms by $x^{m-1} y^n$ (provided $m > 0$) under $T_x$ and $x^m y^{n-1}$ (provided $n > 0$) under $T_y$. For the rhombic Bravais lattice (space group $cmm$), the monomial $x^m y^n$ transforms by $x^{m-1} y^n$ under $T_x$ and a linear combination of $x^{m-1} y^n$ and $x^m y^{n-1}$ (again, assuming $m, n > 0$) under translations parallel to the other lattice direction. In both of these Bravais lattices, including $x^m y^n$ in the multipole group requires both $x^{m-1} y^n$ and $x^m y^{n-1}$ to appear in the multipole group as well. Hence, the same sets of polynomial shift symmetries are allowed in the multipole group as in the monoclinic Bravais lattice. We note, however, that there are now *four* one-dimensional irreps for the orthorhombic lattices. At a given order, there are hence fewer irrep degeneracies that need to be taken into account when considering mixing of irreps under translations than for the monoclinic Bravais lattice.

### 3. Square

The square lattice's point group is $D_4$ [Fig. 1(d)], and its space group is $p4m$. In contrast to the lower-symmetry Bravais lattices, sorting the square lattice polynomials into irreps from first principles requires the use of systematic methods like those in Appendix A. See Table I for the representations under the point group. Mixing of these polynomials under translations is depicted in Fig. 3.

Here, we find a further subtlety in dealing with degenerate representations. The four independent cubic polynomials sort into two copies of the two-dimensional representation $E_1$ of $D_4$. Under translation, the quartic polynomial $A_{--}$ irrep $x^3y + y^3x$ mixes with the cubic $E_1$ irrep $\{x^3 + 3xy^2, y^3 + 3x^2y\}$, while the quartic $A_{+-}$ irrep $x^3y - y^3x$ mixes with the cubic $E_1$ irrep $\{x^3 - 3xy^2, y^3 - 3x^2y\}$. The two degenerate $E_1$ irreps $\{x^3 + 3xy^2, y^3 + 3x^2y\}$ and $\{x^3 - 3xy^2, y^3 - 3x^2y\}$ span all cubic polynomial shift symmetries, so it seems natural to make a tree structure like Fig. 2 using these two $E_1$ irreps as a basis for the cubic polynomial shift symmetries.

However, the quartic $A_{++}$ irrep $x^4 + y^4$ mixes with $\{x^3, y^3\}$, which also transforms as the $E_1$ irrep of $D_4$. We do *not* need to include another cubic irrep of $D_4$. But the irrep $\{x^3, y^3\}$ is a linear combination of the irreps $\{x^3 + 3xy^2, y^3 + 3x^2y\}$ and $\{x^3 - 3xy^2, y^3 - 3x^2y\}$. So the translation mixing here does not respect the simple tree structure we attempted to make.

The conclusion here is that when there are degenerate irreps of a given degree, there is not generally a "canonical" choice of basis for those irreps for which a tree structure like Fig. 2 is unambiguous. For a given set of polynomials that we wish to include in $\mathcal{M}$, compatibility with the space group symmetry requires us to find the "tree" of descendants and include them in $\mathcal{M}$, which is a well-posed problem that can be solved algorithmically. However, that descendant information cannot be neatly encoded for *all* choices of polynomials into a tree of the sort shown in Fig. 2, because different "arrows" in the tree may require different basis choices. One can check that, in the present case, it is not possible to build such a tree, which is why Fig. 3 contains two equivalent bases shown in the shaded circles.

### 4. Triangular

The final Bravais lattice to consider is the triangular lattice. The point group is $D_6$ [Fig. 1(e)] and the space group is $p6m$. There is no additional subtlety compared to the square lattice. See Table II for the results, along with Fig. 4 for the translation mixing tree.

### C. Beyond Bravais: Wallpaper groups

In the presence of a basis for the lattice, several things change compared to a pure Bravais lattice. First, the space group can be any of the 17 wallpaper groups. Second, space group symmetries can in general permute basis sites. Finally, polynomial shift symmetries do not in general need to act in

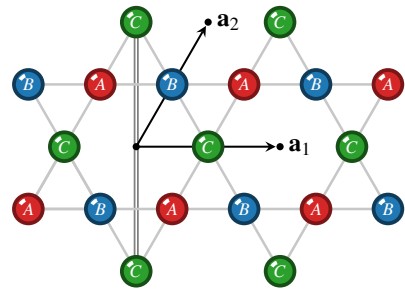

FIG. 5. Kagome lattice with space group $p6m$. $A$, $B$, and $C$ sublattices are labeled in red, blue, and green, respectively, but the sites are identical. The generating mirror, which is chosen to send $x \mapsto -x$, is denoted by the double gray line. The lattice translation vectors $\mathbf{a}_1$ and $\mathbf{a}_2$ are denoted by the black arrows.

the same way on each basis site. The interaction of all of these features leads to significant changes in the possible multipole groups. We will now give a few physically interesting or physically motivated examples which illustrate some key points about multipole groups. In particular, we give examples to illustrate the following facts:

1. The set of allowed multipole groups does not depend only on the space group, and instead depends on the details of the action of the space group on the lattice basis. Using the concept of Wyckoff positions, this action can be classified; we will give a procedure to generate this classification in principle, but will not perform the explicit classification for all 17 wallpaper groups.

2. In the presence of a basis, some multipolar symmetry groups may look very unnatural (particularly those generated by inhomogeneous polynomials), but are actually well-motivated in certain physical contexts.

3. The extended point group may in general be larger than the point group, specifically when the wallpaper group is non-symmorphic.

#### 1. Multipole groups are not a function of space group alone

The kagome and triangular lattices both have space group $p6m$[2]. One might reasonably ask if the allowed multipole groups on the kagome and triangular lattices are "the same."

The naïve answer is immediately "no," simply because there are more polynomial shift symmetries for the kagome lattice than the triangular lattice; on the kagome lattice, one can choose an independent polynomial shift symmetry on each of the three sublattices, whereas there is only one choice of polynomial on the triangular lattice. However, the same argument holds for three layers of the triangular lattice, and it is obvious that one can simply choose an allowed triangular lattice multipole

---

[2] The honeycomb lattice also has space group $p6m$, but we choose to work with the kagome lattice because the results are more generic.

TABLE I. Polynomials up to degree $n = 4$ and their representations under the point group $D_4$ of the square lattice. The irrep labels $A_{\sigma,\nu}$ and $E_k$ are defined explicitly in Appendix A. The descendants of each polynomial under translation mixing are also listed.

| Polynomial(s) | Irrep | Descendants | Polynomial(s) | Irrep | Descendants |
|---|---|---|---|---|---|
| $n = 0$ | | | $n = 3$ | | |
| $1$ | $A_{++}$ | | $\{x^3, y^3\}$ | $E_1$ | $x^2+y^2, x^2-y^2$ |
| | | | $\{x^2 y, y^2 x\}$ | $E_1$ | $x^2+y^2, x^2-y^2, xy$ |
| $n = 1$ | | | $n = 4$ | | |
| $\{x, y\}$ | $E_1$ | $1$ | $x^4+y^4$ | $A_{++}$ | $\{x^3, y^3\}$ |
| $n = 2$ | | | $x^4-y^4$ | $A_{-+}$ | $\{x^3, y^3\}$ |
| $x^2+y^2$ | $A_{++}$ | $\{x, y\}$ | $x^3 y + y^3 x$ | $A_{--}$ | $\{x^3 + 3xy^2, y^3 + 3x^2 y\}$ |
| $x^2-y^2$ | $A_{-+}$ | $\{x, y\}$ | $x^3 y - y^3 x$ | $A_{+-}$ | $\{x^3 - 3xy^2, y^3 - 3x^2 y\}$ |
| $xy$ | $A_{--}$ | $\{x, y\}$ | $x^2 y^2$ | $A_{--}$ | $\{x^2 y, y^2 x\}$ |

TABLE II. Polynomials up to degree $n = 4$ and their representations under the point group $D_6$ of the triangular lattice. We choose the generating mirror $r$ to send $y \mapsto -y$. The irrep labels $A_{\sigma,\nu}$ and $E_k$ are defined explicitly in Appendix A. The descendants of each polynomial under translation mixing are also listed.

| Polynomial(s) | Irrep | Descendants | Polynomial(s) | Irrep | Descendants |
|---|---|---|---|---|---|
| $n = 0$ | | | $n = 3$ | | |
| $1$ | $A_{++}$ | | $x^3 - 3xy^2$ | $A_{-+}$ | $\{x^2-y^2, 2xy\}$ |
| $n = 1$ | | | $y^3 - 3x^2 y$ | $A_{--}$ | $\{x^2-y^2, 2xy\}$ |
| $\{x, y\}$ | $E_1$ | $1$ | $\{x^3+xy^2, y^3+x^2 y\}$ | $E_1$ | $x^2+y^2, \{x^2-y^2, 2xy\}$ |
| $n = 2$ | | | $n = 4$ | | |
| $x^2+y^2$ | $A_{++}$ | $\{x, y\}$ | $(x^2+y^2)^2$ | $A_{++}$ | $\{x^3+xy^2, y^3+x^2 y\}$ |
| $\{x^2-y^2, 2xy\}$ | $E_2$ | $\{x, y\}$ | $\{x^4-y^4, 2(x^3 y+y^3 x)\}$ | $E_2$ | all cubic |
| | | | $\{2(x^3 y-y^3 x), x^4+y^4-6x^2 y^2\}$ | $E_2$ | $x^3-3xy^2, y^3-3x^2 y$ |

group on each of the layers independently (with the space group operations acting simultaneously on all layers). We claim that the kagome lattice does not have this property; there are allowed multipole groups that are fundamentally distinct from all triangular lattice multipole groups. To show this, we classify multipole groups on the kagome lattice.

The wallpaper group $p6m$ is generated by four operations: a six-fold rotation $C$, a reflection $r$, and two translations $T_1$ and $T_2$. With the sublattices labeled as in Fig. 5, space group operations act as

$$C \begin{cases} \phi_A(\mathbf{r}) \\ \phi_B(\mathbf{r}) \\ \phi_C(\mathbf{r}) \end{cases} = \begin{cases} \phi_C(M_C \mathbf{r}) \\ \phi_A(M_C \mathbf{r}) \\ \phi_B(M_C \mathbf{r}) \end{cases}, \qquad r \begin{cases} \phi_A(\mathbf{r}) \\ \phi_B(\mathbf{r}) \\ \phi_C(\mathbf{r}) \end{cases} = \begin{cases} \phi_B(M_r \mathbf{r}) \\ \phi_A(M_r \mathbf{r}) \\ \phi_C(M_r \mathbf{r}) \end{cases} \quad (8)$$

$$T_i \phi_a(\mathbf{r}) = \phi_a(\mathbf{r} + \mathbf{t}_i) \quad (9)$$

where the six-fold-rotation and reflection matrices, $M_C$ and $M_r$, respectively, and the two translation vectors, are given by

$$M_C = \begin{pmatrix} \frac{1}{2} & -\frac{\sqrt{3}}{2} \\ \frac{\sqrt{3}}{2} & \frac{1}{2} \end{pmatrix}, \qquad M_r = \begin{pmatrix} -1 & 0 \\ 0 & 1 \end{pmatrix} \quad (10)$$

$$\mathbf{t}_1 = a(1, 0), \qquad \mathbf{t}_2 = a\left(\frac{\sqrt{3}}{2}, \frac{1}{2}\right) \quad (11)$$

with $a$ the lattice constant. The point group is $D_6$, the symmetries of a regular hexagon. The polynomial shift symmetry

irreps of the point group are given in Table III, and the translation dependences are shown in Fig. 6.

Indeed, every triangular lattice multipole group has a corresponding kagome lattice multipole group, which acts identically on all three sublattices. However, even constant polynomials show new features. There are constant polynomials that form a 2D representation of the point group, which cannot happen on the triangular lattice or several copies of the triangular lattice. There are therefore only four possible multipole groups with only constant polynomial shift symmetries on the kagome lattice; there are independent choices of whether to include each of the two irreps, but those are the only possibilities. Compare this to three copies of the triangular lattice, where any constant shift $f = (a, b, c)^T$ is a polynomial shift symmetry consistent with the space group. Physically, this is very interesting; including $\{v_x, v_y\}$ but not $v_0$ means that conserving a vector charge but *not* the total scalar charge is consistent with the point group symmetry. We discuss the physical implications of this fact further in Sec. IV D. Similarly, the "dependency tree" in Fig. 6 does not decouple into multiple copies of the triangular lattice tree.

We conclude that, while the kagome lattice admits multipole groups identical to those of the triangular Bravais lattice, it also allows multipole groups that are fundamentally different from any multipole group on the Bravais lattice. The allowed multipole groups are therefore not simply a function of the space group.

TABLE III. Polynomials up to degree $n = 2$ and their representations under the point group $D_6$ of the kagome lattice. We choose the generating mirror $r$ to send $x \mapsto -x$, and use the notation $v_0 = (1, 1, 1)^T$, $v_x = \frac{1}{2}(\sqrt{3}, -\sqrt{3}, 0)^T$, and $v_y = \frac{1}{2}(-1, -1, 2)^T$. For the *breathing* kagome lattice (for which the point group is $D_3$), the irrep labels are modified according to Eq. (16), but the polynomials and their descendants remain unmodified.

| Polynomial(s) | Irrep | Descendants | Polynomial(s) | Irrep | Descendants |
|---|---|---|---|---|---|
| **$n = 0$** | | | **$n = 2$** | | |
| $v_0$ | $A_{++}$ | | $(x^2 + y^2)v_0$ | $A_{++}$ | $\{x, y\}v_0$ |
| $\{v_x, v_y\}$ | $E_2$ | | $\{2xy, (x^2 - y^2)\}v_0$ | $E_2$ | $\{x, y\}v_0$ |
| **$n = 1$** | | | | | |
| $xv_x + yv_y$ | $A_{-+}$ | $\{v_x, v_y\}$ | $-(x^2 - y^2)v_x + 2xyv_y$ | $A_{+-}$ | $\{yv_x + xv_y, xv_x - yv_y\}$ |
| $yv_x - xv_y$ | $A_{--}$ | $\{v_x, v_y\}$ | $(x^2 - y^2)v_y + 2xyv_x$ | $A_{++}$ | $\{yv_x + xv_y, xv_x - yv_y\}$ |
| $\{x, y\}v_0$ | $E_1$ | $v_0$ | $\{v_x, -v_y\}(x^2 + y^2)$ | $E_2$ | all linear |
| $\{yv_x + xv_y, xv_x - yv_y\}$ | $E_1$ | $\{v_x, v_y\}$ | $\{2xyv_y + (x^2 - y^2)v_x, -2xyv_x + (x^2 - y^2)v_y\}$ | $E_2$ | $(x^2 - y^2)v_y + 2xyv_x, -(x^2 - y^2)v_x + 2xyv_y$ |

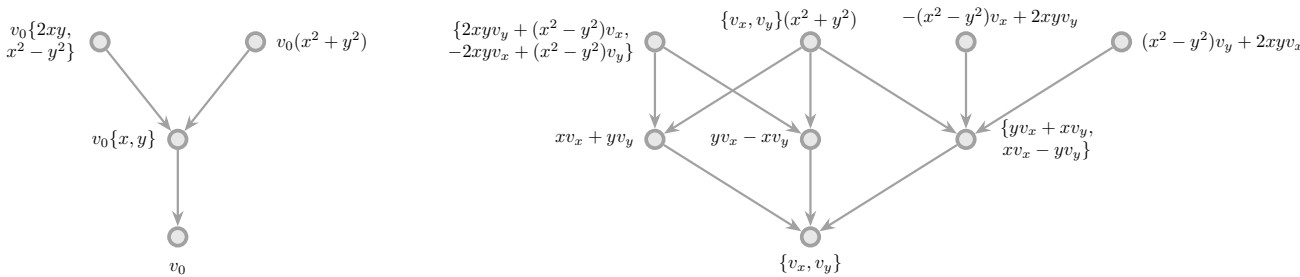

FIG. 6. Mixing of irreps by translational symmetry for the kagome (and breathing kagome) lattice(s). Every arrow means that some (infinitesimal) translation acting on the polynomial(s) of higher degree generates the polynomial(s) of lower degree. Unlike previous trees, there exist two disjoint hierarchical structures. Physically, this means that there exist valid multipole groups that do not include total charge. The shorthand $v_0 = (1, 1, 1)^T$, $v_x = \frac{1}{2}(\sqrt{3}, -\sqrt{3}, 0)^T$, and $v_y = \frac{1}{2}(-1, -1, 2)^T$ is used for the vectors that span the three-dimensional space that corresponds to the sublattice index. The representations of the polynomials under the point group are given in Table III.

### 2. A classification procedure

We saw above that the way that space group symmetries permute different basis sites in the lattice can substantially change the structure of the multipole group, and so the space group alone does not determine the allowed multipole groups. For a given space group, any set of basis sites can be decomposed into sets of independent, decoupled symmetry orbits; each orbit type is called a Wyckoff position, and the number of atoms in the orbit is called the multiplicity of the Wyckoff position. Wyckoff positions have been classified exhaustively for all space groups in both 2D and 3D [46–51]. Since space group operations never mix lattice sites with different Wyckoff positions, one can independently choose allowed polynomial shift symmetries for each Wyckoff position.

Given a crystalline lattice, then, we can classify all possible multipole groups as follows. First, identify the space group of the crystalline lattice. Then, decompose the set of atoms in the unit cell into distinct Wyckoff positions. Next, for each Wyckoff position, compute the allowed multipole groups as we did above. A multipole group is generated by the space group operations and various polynomial shift symmetries

that transform as direct sums of irreps of the extended point group, one summand per Wyckoff position. If a polynomial shift symmetry appears in $\mathcal{M}$, so must its "descendants" under translations, which will be direct sums of the descendants of each individual irrep summand.

Given a Wyckoff position, the problem of sorting polynomial shift symmetries into irreps of the extended point group can be broken down further. First, one can determine the irrep of the extended point group formed by the permutation action of the space group on the basis sites. This is purely geometric information determined completely by the Wyckoff position. Second, one can take pure polynomials, without any reference to the Wyckoff position, and sort them into irreps of the extended point group. Finally, the irreps obtained in the prior two steps are combined using the Clebsch-Gordon coefficients of the extended point group; see Appendix A for details.

As an example of the above procedure, we can consider the classification problem for the wallpaper group *pm*. Assuming, without loss of generality, that the point group $D_1 \cong \mathbb{Z}_2$ is generated by reflections $x \leftrightarrow -x$, there are three Wyckoff positions, shown in Fig. 7a. Wyckoff position $a$ is an atom placed at $(0, y)$ for any $y$. Wyckoff position $b$ is an atom at

$(1/2, y)$ for any $y$. Wyckoff position $c$ consists of a pair of atoms at $(\pm x, y)$ for any $y$ (here, $x$ is interpreted modulo the lattice constant in the $x$ direction).

For atoms on either the $a$ and $b$ Wyckoff positions, polynomial shift symmetries do not permute basis sites within the unit cell. The multipole group classification for these atoms can be read off by inspection. The monomial $x^m y^n$ forms a trivial (resp. nontrivial) irrep of $\mathbb{Z}_2$ if $m$ is even (resp. odd), and translations mix irreps in the same way as Fig. 2.

For atoms on a $c$ Wyckoff position, the point group operation $r$ interchanges the two basis elements, that is,

$$r \begin{pmatrix} \phi_A(\mathbf{r}) \\ \phi_B(\mathbf{r}) \end{pmatrix} \rightarrow \begin{pmatrix} \phi_B(M_r \mathbf{r}) \\ \phi_A(M_r \mathbf{r}) \end{pmatrix}. \tag{12}$$

It is straightforward to see that irreps of the point group are given by transformations

$$\begin{pmatrix} \phi_A(\mathbf{r}) \\ \phi_B(\mathbf{r}) \end{pmatrix} \rightarrow \begin{pmatrix} \phi_A(\mathbf{r}) \\ \phi_B(\mathbf{r}) \end{pmatrix} + x^m y^n v_k, \tag{13}$$

with $v_0 = (1, 1)^T$ and $v_1 = (1, -1)^T$. The irrep is trivial if $m + k$ is even and nontrivial if $m + k$ is odd. Note that the permutation action on the basis sites is a direct sum of a trivial irrep $v_0$ and a nontrivial irrep $v_1$, so the polynomial shift symmetries are just the (tensor) product of one of these $v_k$ and the irrep formed by polynomials $x^m y^n$ in the spatial coordinates. The corresponding Clebsch-Gordon coefficients are trivial because the irreps involved are all one-dimensional.

For a generic arrangement of atoms with space group $pm$, then, we choose a set of polynomial shift symmetries that we wish to include. Each shift symmetry should be decomposed into a linear combination of direct sums of irreps of the extended point group, with one (possibly zero) summand in the direct sum for each Wyckoff position in the lattice. The complete symmetry orbit of the original polynomial shift symmetries then follows from the symmetry orbit and translation mixing of each direct summand.

For example, suppose we have atoms at Wyckoff position $a$ (call the corresponding field $\phi^{(a)}$) and Wyckoff position $c$ (call the corresponding fields $\phi^{(c)}_{A,B}$). We could choose to include the shift symmetry that transforms the $a$ Wyckoff position by $x^2 y$ (trivial irrep of $\mathbb{Z}_2$) and transforms the $c$ Wyckoff position by $\lambda y v_1$ (nontrivial irrep of $\mathbb{Z}_2$), where $\lambda$ is some constant with dimensions of length$^2$:

$$\begin{pmatrix} \phi^{(a)}(\mathbf{r}) \\ \phi^{(c)}_A(\mathbf{r}) \\ \phi^{(c)}_B(\mathbf{r}) \end{pmatrix} \rightarrow \begin{pmatrix} \phi^{(a)}(\mathbf{r}) \\ \phi^{(c)}_A(\mathbf{r}) \\ \phi^{(c)}_B(\mathbf{r}) \end{pmatrix} + \begin{pmatrix} x^2 y \\ \lambda y \\ -\lambda y \end{pmatrix}. \tag{14}$$

Under a mirror (as shown in Fig. 7a)

$$r \begin{pmatrix} x^2 y \\ \lambda y \\ -\lambda y \end{pmatrix} \rightarrow \begin{pmatrix} x^2 y \\ -\lambda y \\ \lambda y \end{pmatrix}, \tag{15}$$

since the first entry forms a trivial irrep of $\mathbb{Z}_2$ and the last two entries form a nontrivial irrep of $\mathbb{Z}_2$. Hence, the multipole group

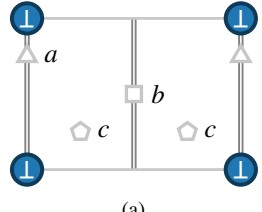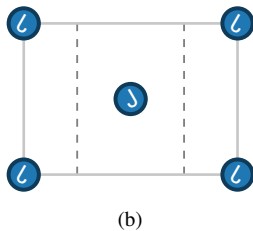

(a)          (b)

FIG. 7. Left: An example of a lattice belonging to the space group $pm$. Axes of reflection are denoted by the double gray lines. The labels $a$, $b$, and $c$ correspond to the three types of Wyckoff position, with multiplicities one, one, and two, respectively. Right: A lattice with space group $pg$. The dashed gray lines represent axes of glide reflection since the two types of lattice site are (only) mapped onto one another under the mirror $x \mapsto -x$ (supposing they were spatially coincident). The solid gray lines represent lattice translations, which are parallel to the $x$- and $y$-axes.

must also contain $(x^2 y, -\lambda y, \lambda y)^T$. We know the translation descendants of each irrep ($x^2 y$ on the $a$ Wyckoff position descends to $xy, x, y, 1$ and $y v_1$ on the $c$ Wyckoff position descends to $v_1$), but we must translate both irreps simultaneously to get the descendants of the combination. Namely, $(x^2, \pm \lambda, \mp \lambda)^T$ are descendants, but $(x^2, 0, 0)^T$ is not a descendant.

We comment that the difference between the triangular, multi-layer triangular, honeycomb, and kagome lattices are exactly which Wyckoff positions are occupied; all of them have space group $p6m$. According to the nomenclature in the Bilbao crystallographic server [47–49], the triangular lattice consists of an atom at the $a$ Wyckoff position, and the multi-layer triangular lattice consists of multiple atoms at the $a$ Wyckoff position. The honeycomb lattice has atoms in the $b$ Wyckoff position (which has multiplicity two), and the kagome lattice has atoms in the $c$ Wyckoff position (multiplicity three). Space group $p6m$ also has two multiplicity-six Wyckoff positions, occupying one of which would generate the ruby lattice, and a multiplicity-12 Wyckoff position; the classification procedure for these Wyckoff positions using the techniques developed herein is straightforward but tedious.

### 3. Scope of the classification and physical interpretation of symmetries

There is an important subtlety in the notation when we write Eq. (2). In the absence of basis sites, there is no $i$ index, and in the continuum, it is physically sensible to restrict our attention to continuous $f(\mathbf{r})$. On the lattice, however, there is no *a priori* physical reason to restrict to continuous $f_i(\mathbf{r})$. Rather than writing something like Eq. (2), we could instead consider shift symmetries where the field in question is shifted by an arbitrary number chosen independently at each basis site. This is the most general possibility, but attempting to classify such functions in any useful way is beyond the scope of this paper.

One natural way to restrict our attention to polynomial shift symmetries is to imagine defining a polynomial shift symmetry $f(\mathbf{r})$ on the entire plane (without reference to basis sites), and

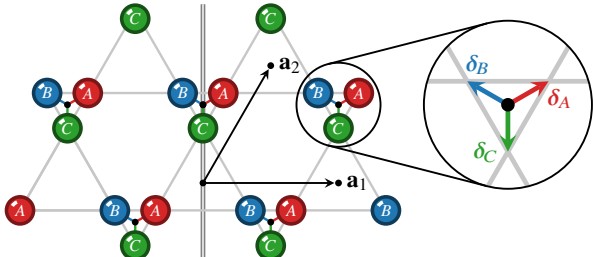

FIG. 8. Breathing kagome lattice, which belongs to the space group $p3m1$. The $A$, $B$, and $C$ sublattices are labeled in red, blue, and green, respectively, but the sites are identical. The generating mirror, which is chosen to send $x \mapsto -x$, is denoted by the double gray line. The lattice translation vectors $\mathbf{a}_1$ and $\mathbf{a}_2$ are denoted by the black arrows. The $C_3$ rotation center to which each basis site is associated is depicted by a solid black circle. The zoomed circular inset illustrates the three vectors $\boldsymbol{\delta}_i$ that connect Bravais sites to basis sites.

then defining a shift symmetry on the lattice by restricting the domain of $f(\mathbf{r})$ to $\mathbf{r}$ that belong to the crystalline lattice. This is a completely reasonable possibility, and it is included in our formalism. However, one could also define one polynomial $f_i(\mathbf{r})$ per basis site, and then defining a shift symmetry on the lattice by restricting the domain of $f_i(\mathbf{r})$ to those $\mathbf{r}$ that are an $i$th basis element. The latter is the approach we take in this paper because it is more general; for a generic lattice with two basis elements $A$ and $B$, one could not obtain, e.g., $f_A(\mathbf{r}) = x$ and $f_B(\mathbf{r}) = y$ by restricting the same finite-order polynomial to both the $A$ and $B$ sublattices.

This subtlety in notation can sometimes obscure the physical meaning of certain multipolar symmetries. As an example which will become relevant in Sec. IV D, we consider the classification of multipole groups on the *breathing* kagome lattice, shown in Fig. 8. The breathing kagome lattice has space group $p3m1$, with point group $D_3$. The allowed multipole groups for the breathing kagome lattice are almost identical to those for the kagome lattice. One can check that the table for breathing kagome can be obtained from Table III by simply replacing the irrep labels:

$$E_{1,2}\big|_{D_3} = E_1, \qquad A_{\sigma,\nu}\big|_{D_3} = A_{+,\nu} \qquad (16)$$

with identical translation mixing. The only difference in the allowed multipole groups is that some irreps that are non-degenerate on the kagome lattice become degenerate on the breathing kagome lattice, which allows us to take linear combinations of these newly degenerate irreps without including both individual irreps. For example, $(x^2 + y^2)v_0$ transforms as $A_{++}$ on both the kagome and breathing kagome lattice, while $xv_x + yv_y$ transforms as $A_{-+}$ on kagome and $A_{++}$ on breathing kagome. On the kagome lattice, then, including $(x^2 + y^2)v_0 + (xv_x + yv_y)$ forces both $(x^2 + y^2)v_0$ and $(xv_x + yv_y)$ to appear in the multipole group, along with their descendants $\{x, y\}v_0$, $v_0$, and $\{v_x, v_y\}$. However, on the breathing kagome lattice, $(x^2 + y^2)v_0 + (xv_x + yv_y)$ transforms trivially under the point group, so only its descendants $\{2xv_0 + v_x, 2yv_0 + v_y\}$ and $v_0$ need to appear in the multipole group.

Now consider the following polynomial shift symmetry:

$$\begin{pmatrix} \phi_A(\mathbf{r}) \\ \phi_B(\mathbf{r}) \\ \phi_C(\mathbf{r}) \end{pmatrix} \rightarrow \begin{pmatrix} \phi_A(\mathbf{r}) \\ \phi_B(\mathbf{r}) \\ \phi_C(\mathbf{r}) \end{pmatrix} + \begin{pmatrix} x - \boldsymbol{\delta}_A \cdot \hat{\mathbf{x}} \\ x - \boldsymbol{\delta}_B \cdot \hat{\mathbf{x}} \\ x - \boldsymbol{\delta}_C \cdot \hat{\mathbf{x}} \end{pmatrix} \qquad (17)$$

where $\boldsymbol{\delta}_i$ is the displacement of the $i$th basis site from an associated Bravais lattice site, as shown in Fig. 8, and the $\boldsymbol{\delta}_i$ transform as vectors under symmetry operations. This shift symmetry is not, *a priori*, obviously physically meaningful. However, by inspection we can see that this amounts to shifting each field by $\mathbf{R} \cdot \hat{\mathbf{x}}$, where $\mathbf{R}$ is the closest Bravais lattice site. Physically, the presence of this symmetry means that the $x$ component of the total dipole moment is conserved, where each field's charge is weighted by the closest Bravais lattice site's position instead of the physical position of the charge itself. One situation where this is physically meaningful is as follows. Imagine placing a spin-3/2 $f$-like electron at the center of each small triangle and a localized $d$ electron at each vertex of the breathing kagome lattice. With a strong Ising-like interaction between the $d$ and $f$ electrons of the sort

$$\hat{H}_\triangledown = -\hat{S}_z^{(f)} \left( \hat{S}_z^{(d,A)} + \hat{S}_z^{(d,B)} + \hat{S}_z^{(d,C)} \right) \qquad (18)$$

on each triangle, one could imagine that the $f$ electron's spin is equal to the sum of the surrounding $d$ electron spins. When allowing $d$ electrons to weakly interact between triangles, the symmetry Eq. (17) means that (a component of) the dipole moment associated to the $f$ electrons is conserved.

#### 4. Nonsymmorphic symmetries

When the space group is nonsymmorphic, namely when the space group is not a direct product of a point group and translations, the extended point group becomes distinct from the point group.

As an example, consider the space group $pg$, generated by translations and a glide consisting of reflection about the $x$-axis (without loss of generality) combined with a half-translation. An example lattice and its symmetries is given in Fig. 7b, where the two types of lattice site are assumed to transform into each other under reflections about the $x$-axis only.

Using the same formalism and applying a glide, we obtain

$$\phi_{A,B}(\mathbf{x}) \rightarrow \phi_{B,A}\left( G\mathbf{x} + \tfrac{1}{2}(\mathbf{a}_1 + \mathbf{a}_2) \right) \qquad (19)$$

where $G = \text{diag}(1, -1)$. Now there is a nontrivial matrix acting on the coordinates rather than simply a translation. The same formalism as earlier applies, but now instead of finding irreps of the point group, we should find irreps of the extended point group $\mathbb{Z}_2$, which is generated by the transformation

$$\phi_{A,B}(\mathbf{x}) \rightarrow \phi_{B,A}(G\mathbf{x}). \qquad (20)$$

This extended point group is isomorphic as a group to $S/T$ where $S$ is the space group and $T \subset S$ is the subgroup consisting of pure translations.

It is not hard to check that the irreps of the extended point group are $x^m y^n v_k$, where $v_k = (1, (-1)^k)^T$ with $k = 0, 1$. The irrep is trivial if $n + k$ is even and nontrivial if $n + k$ is odd. After incorporating translations, it is straightforward to check that including $x^m y^n v_k$ in the multipole group forces us to include $x^{m'} y^{n'} v_k$ for any $0 \leq m' < m$, $0 \leq n' \leq n$. These irreps form two decoupled copies of the tree in Fig. 2, one for each $v_k$. The formalism we have developed is therefore capable of constructing the allowed multipole groups to any desired order on any lattice.

### III. CONSTRUCTING MODELS

Given a crystalline multipole group $\mathcal{M}$, one would generally wish to construct models, both in the continuum and on the lattice, with $\mathcal{M}$ symmetry in order to study the effect of these exotic symmetries on interesting properties like their dynamics. In fact, as we will describe shortly, there is reason to expect that some multipolar symmetries, particularly when the multipole group is sub-maximal, can lead to highly interesting consequences like a sub-extensive number of emergent conserved quantities at long wavelengths. Such models are also of interest to guide the search for emergent multipolar symmetries in real materials. In this section we explain how one may construct effective Hamiltonians and field theories given a multipolar group $\mathcal{M}$.

#### A. Spin models

We can also view the polynomials that comprise the multipole groups $\mathcal{M}$ that we have constructed as corresponding to conserved multipole moments of a local charge density. The construction of the multipole group ensures that we are able to write down Hamiltonians that are invariant under the space group of the lattice and conserve these multipole moments. Specifically, given spin-$S$ degrees of freedom living on the sites of a lattice $\mathcal{L}$, we would like to construct tranlation invariant Hamiltonians that conserve the multipole moments

$$\hat{\mathcal{Q}}[f] = \sum_{i \in \mathcal{L}} f(\mathbf{r}_i) \hat{S}_i^z, \tag{21}$$

of the local "charge density" $\hat{S}_i^z$ for all polynomials $f(\mathbf{r})$ belonging to the multipole group $\mathcal{M}$. As we discussed in Sec. II C 3, in the presence of a basis, the weighting function $f(\mathbf{r})$ corresponds to an in-principle independent multipole moment on each basis site.

We now describe a systematic way to construct local Hamiltonians that conserve the moments (21) (with further details provided in Appendix D). The Hamiltonians are composed of local "gates" $\hat{h}_{\mathbf{x}\alpha}$ and their Hermitian conjugates. Gates are labeled by the index $\alpha$ (there may be more than one type of gate compatible with the imposed conservation laws) and are centered on $\mathbf{x}$ (which may or may not coincide with one of the basis sites). Hamiltonians built from such gates take the

general form

$$\hat{H} = \sum_{\mathbf{x}, \alpha} g_\alpha (\hat{h}_{\mathbf{x}\alpha} + \hat{h}_{\mathbf{x}\alpha}^\dagger), \tag{22}$$

where the $g_\alpha$ are coupling constants. The gates $\hat{h}_{\mathbf{x}\alpha}$ are local, acting on spins belonging to a 'cluster' of spins $C \subset \mathcal{L}$ by incrementing or decrementing the $z$ component of the spins by integers $n_\alpha(\boldsymbol{\delta})$ in the vicinity of position $\mathbf{x}$:

$$\hat{h}_{\mathbf{x}\alpha} = \prod_{i \in C} \left( \hat{S}_{\mathbf{x}+\boldsymbol{\delta}_i}^{\mathrm{sgn}(n_\alpha(\boldsymbol{\delta}_i))} \right)^{|n_\alpha(\boldsymbol{\delta}_i)|}. \tag{23}$$

We will work exclusively with clusters of strictly finite support. This generic gate structure, or a specific variant thereof, has previously been utilized in Refs. [13, 29, 40–42] to construct Hamiltonians that conserve various moments of $\hat{S}_i^z$. When evaluating the time evolution of the charges (21) through their Heisenberg equation of motion, $i\partial_t \hat{\mathcal{Q}}[f] = [\hat{\mathcal{Q}}[f], \hat{H}]$, the coefficients $n_\alpha(\boldsymbol{\delta})$ that define the gate effect a discrete derivative $D_\alpha$ acting on the function $f$. Importantly, the charge $\hat{\mathcal{Q}}[f]$ will be a conserved quantity if the derivative $D_\alpha$ annihilates $f$ for all $\mathbf{x}$, with this property being preserved under arbitrary perturbations (or operator insertions in $\hat{h}_{\mathbf{x}\alpha}$, see Appendix D) that are diagonal in the $\hat{S}_i^z$ basis. Hence, we can systematically construct Hamiltonians of the form (22) that conserve a finite list of multipole moments $f(\mathbf{r})$ that form a multipole group $\mathcal{M}$ by identifying the possible discrete derivatives, specified by $\{n_\alpha(\boldsymbol{\delta}_i)\}$, that annihilate all $f(\mathbf{r})$, i.e., $D_\alpha f = 0$. Producing a Hamiltonian that is *invariant* under the space group from a given discrete derivative requires an extra step: One must find the orbit of the discrete derivative under the action of the space group and include *all* such operators as "gates" in Eq. (22) with equal weight $g_\alpha$ (all such operators generated in this manner from a valid discrete derivative are guaranteed to annihilate all polynomials belonging to the multipole group). In this way, space group operations will then just permute the gates, leaving $\hat{H}$ unchanged.

#### 1. Constructing discrete derivatives

For a given multipole group, there are two handles that we can use to constrain the search for possible discrete derivatives: (i) the local Hilbert space dimension through the spin, $S$, and (ii) the size (and shape) of the cluster $C$. The size of the spin directly constrains the maximum absolute integer change in $\hat{S}_i^z$ to be $\leq 2S$, while reducing the size of the cluster $C$ reduces the dimension of the parameter space for the search; the total number of gates within a region of size $|C|$ is $(2S + 1)^{|C|}$.

For convenience, we arrange the integers that define the gate into a 'vector' $|n_\alpha\rangle$, where $0 \leq |n_\alpha| \leq 2S$. In this language, a judicious choice of origin[3] allows us to write the action of the

-----

[3] The group property of the multipole group allows us to shift the 'origin' at will: If $(f | n_\alpha) = 0$ for all $f \in \mathcal{M}$, then $\sum_{i \in C} f(\mathbf{r}_i + \mathbf{t}) n_\alpha(\mathbf{r}_i) = 0$ for all $\mathbf{t}$ since $f(\mathbf{r}_i + \mathbf{t})$ will generate a linear combination of polynomials belonging to $\mathcal{M}$.

discrete derivative as

$$(D_\alpha f)(\mathbf{0}) = \sum_{i \in C} f(\boldsymbol{\delta}_i) n_\alpha(\boldsymbol{\delta}_i) \equiv (f|n_\alpha) \stackrel{!}{=} 0\,, \qquad (24)$$

while the group status of $\mathcal{M}$ ensures that $(D_\alpha f)(\mathbf{x})$ will vanish for all $\mathbf{x}$. Equation (24) therefore states that any gate that is compatible with all conservation laws, specified by the integer vector $|n_\alpha)$, must be orthogonal to all vectors $|f) \in \mathbb{R}^{|C|}$ corresponding to polynomials that belong to the multipole group. To touch base with continuum derivative notation natural for field theoretic treatments, we can find the continuum derivative to which $D_\alpha$ coarse grains by writing

$$(D_\alpha g)(\mathbf{x}) = \sum_{i \in C} n_\alpha(\boldsymbol{\delta}_i) g(\mathbf{x} + \boldsymbol{\delta}_i)$$
$$= \sum_n \sum_{m=0}^n \frac{1}{n!}\binom{n}{m}(x^{n-m}y^m|n_\alpha)\partial_x^{n-m}\partial_y^m g(\mathbf{x}) \quad (25)$$

where, in the second line, we performed a Taylor expansion of the infinitely differentiable function $g(\mathbf{x})$ and replaced the sum over sites in the cluster with an inner product according to Eq. (24). Hence, to find the overlap of $D_\alpha$ with continuum derivatives of the form $\partial_x^m \partial_y^n$, one simply has to evaluate the overlap between the vector $|n_\alpha)$ that defines the discrete derivative and the vectorized monomial $x^m y^n$ (weighted by the appropriate combinatorial factor). Note that the reverse process – finding discrete derivatives from continuum derivatives – will *not* in general produce valid discrete derivatives that annihilate all $f$. This failure can occur when the multipole group is sub-maximal and a derivative of order $m$, strictly less than the maximum polynomial degree $n$, annihilates all polynomials belonging to $\mathcal{M}$. Requiring that the discrete derivative coarse grains to the order-$m$ derivative does not constrain its overlap with derivatives of order $\ell > m$. This spurious overlap with derivatives of order $\ell$ can then prevent the order-$\ell$ polynomials from being annihilated by the discrete derivative. Indeed, this subtlety will be responsible for the interesting physical consequences that we discuss in Sec. IV. One should therefore always solve Eq. (24) in order to find the correct discrete derivative operators.

## IV. PHYSICAL CONSEQUENCES

We have explained how the possible multipolar groups consistent with lattice symmetries may be constructed on arbitrary lattices. We now illustrate the power of the above formalism by exploring some physical consequences of multipolar symmetries. To this end, we pick one striking phenomenon that has previously been discussed, and generalize it to arbitrary crystal lattices, and also identify two remarkable phenomena that have not previously been discussed, as far as we are aware, but which we encounter when we explore multipolar groups on triangular and breathing kagome lattices respectively. A common theme in many of these discussions is the emergence of a (generally sub-extensive) set of conserved quantities at sufficiently long wavelengths for certain sub-maximal multipole

groups. We work in the continuum to motivate the existence of these conserved quantities, and then identify examples with analogous behavior on lattices that host degrees of freedom with strictly finite local Hilbert space dimensions, subject to locality requirements.

### A. General considerations – emergent symmetries

Given a scalar field $\phi$ (for simplicity) subject to some polynomial shift symmetries, the field $\phi$ can only appear in a symmetry-respecting Lagrangian as $\mathcal{D}_\alpha \phi$, where $\mathcal{D}_\alpha$ is some derivative operator that obeys

$$\mathcal{D}_\alpha f(\mathbf{x}) = 0 \qquad (26)$$

for all polynomials $f(\mathbf{x})$ in the multipole group. The notation $\mathcal{D}_\alpha$ is reserved for continuum derivatives, in order to distinguish them from their discrete counterparts $D_\alpha$. Consider a multipole group whose highest-degree polynomials are degree $n$; then any $\mathcal{D}_\alpha$ of order $(n+1)$ or higher will solve the above equation. However, if the multipole group is sub-maximal, there may exist some $\mathcal{D}_\alpha$ of order $m \leq n$ that annihilate all polynomials belonging to the group. As we will see, in some cases it may happen that the lowest-order $\mathcal{D}_\alpha$ annihilate additional polynomials $g(\mathbf{x})$ that are not in the multipole group, although higher-order solutions will generally not annihilate the additional polynomials. In this case, the additional polynomials lead to additional quantities

$$\mathcal{Q}[g] = \int d^2\mathbf{x}\, g(\mathbf{x})\rho(\mathbf{x})\,, \qquad (27)$$

where $\rho(\mathbf{x})$ is the density operator of the scalar field, which are (emergently) conserved at long wavelengths.

One of the simplest examples of this phenomenon is a theory on the square lattice (see Sec. II B 3) that exhibits the following polynomial shift symmetries (see Ref. [42]):

$$f(\mathbf{x}) \in \{1,\, x,\, y,\, xy,\, x^2 - y^2\}\,. \qquad (28)$$

This is a valid multipole group; the polynomials $x^2 - y^2$ and $2xy$ form distinct one-dimensional irreps of the point group $D_4$ ($A_{-+}$ and $A_{--}$, respectively), and translation invariance requires that charge and both components of dipole are also conserved (see the hierarchy in Fig. 3). While this list of functions is annihilated by any third-order generalized derivative, all functions are also annihilated by the two-dimensional Laplacian $\nabla^2 = \partial_x^2 + \partial_y^2$, which is second order in derivatives and the only such operator. However, we may immediately observe that any harmonic function $g(\mathbf{x})$, i.e., $\nabla^2 g = 0$, will also be annihilated by the lowest-order generalized derivative $\nabla^2$. Each such harmonic function $g$ defines a quasi-conserved quantity via Eq. (27). This infinite family of conservation laws is broken by higher-order, dangerously irrelevant derivative corrections that cease to annihilate the harmonic functions.

## B. Localization in discrete Laplacian models

It was recently shown in Ref. [29] that if the dynamics on a generic lattice is given entirely by certain "discrete Laplacian" operators[4], then the system exhibits strong fragmentation leading to localization [12, 13]. In our language, such gates are of the form $n_{\partial^2}(\mathbf{0}) = z$ and $n_{\partial^2}(\boldsymbol{\delta}) = -1$ for all vectors $\boldsymbol{\delta}$ that connect a site to its $z$ nearest neighbors (and the gate with the signs of $n_{\partial^2}$ flipped). We will see examples of these gates momentarily. Using the formalism developed thus far, we are able to write down minimal valid multipole groups for which discrete Laplacians are the smallest allowed gates. Therefore, if the gate sizes are restricted to be the size of the discrete Laplacian and smaller, these symmetries are sufficient to enforce strong shattering/fragmentation in two dimensions. We illustrate this by explicitly working out the minimal necessary multipole group on the square and triangular lattices, although our formalism could obviously be extended to obtain the minimal sufficient multipole group on any lattice, in arbitrary dimensions.

### 1. Square lattice

For the square lattice, the minimal set of conserved multipole moments that is required to give rise to localization and is compatible with space group symmetry is:

$$f(\mathbf{r}) = \{1, x, y, xy, x^2 - y^2\}, \qquad (29)$$

which produces the discrete Laplacian gate

$$(30)$$

as the *unique* solution with smallest range. In (30), the color of a site denotes the sign of $n(\boldsymbol{\delta}_i)$, and the integer determines $|n(\boldsymbol{\delta}_i)|$, the combination of which determines the gate through Eq. (23). Since $xy$ and $x^2 - y^2$ transform as one-dimensional irreps of $D_4$, we are able to remove just one of them whilst maintaining a valid multipole group. However, removing $xy$ allows gates that correspond to discretizations of $\partial_x^2$ and $\partial_y^2$ separately. On the other hand, removing $x^2 - y^2$ permits lattice discretizations of $\partial_x \partial_y$, which can be discretized on the sites that surround a single plaquette, permitting operators with smaller range. The linear-order polynomials are not required to eliminate the undesired gates, but are required by translation symmetry.

---

[4] While Ref. [29] does not require two dimensions or a translation-invariant lattice, we will specialize to crystal structures that satisfy these requirements in the discussion that follows.

### 2. Triangular lattice

On the triangular lattice, the set of multipole moments (29) is insufficient to fully constrain the gate of smallest range to be uniquely determined; instead, there are two $D_3$-symmetric solutions with $n(\mathbf{0}) = z/2$. This can be remedied by including an additional polynomial in the multipole group:

$$f(\mathbf{r}) = \{1, x, y, xy, x^2 - y^2, x^3 - 3xy^2\}, \qquad (31)$$

which leads to the desired discrete Laplacian gate

$$(32)$$

as the unique solution. The polynomial $y^3 - 3x^2y$ can also be included in the multipole group (31) without changing the fundamental solution (32). If $y^3 - 3x^2y$ is instead the *only* third-order polynomial in the multipole group, then the allowed gates are no longer unique; the same two solutions are permitted as for the set of multipole moments in Eq. (29).

## C. Subsystem symmetries in 2D from $O(1)$ symmetries

We now discuss the first of the apparently new phenomena that we discover by applying our formalism, in this case to particular multipole groups on the triangular lattice. In particular, we find that, for certain choices of the multipole group $\mathcal{M}$, all gates up to a certain size – parametrized by discrete derivatives – will additionally conserve (nonorthogonal) subsystem symmetries along lines of the triangular lattice. In this way, if we restrict the support of the possible gates, a finite list of conserved multipole moments on the triangular lattice will give rise to a much stronger emergent subsystem symmetry, involving an $O(L)$ number of emergent conserved quantities for a lattice of linear dimension $L$.

We begin by considering the fourth-order polynomial $f(\mathbf{x}) = (x^2 + y^2)^2$, which transforms according to the trivial representation, $A_{++}$, of the point group $D_6$ (see Table II). The 'descendant' polynomials of lower order $k < 4$ that must additionally be included for $\mathcal{M}$ to be closed can be found in Fig. 4. We further suppose that the third-order polynomial $f(\mathbf{x}) = y^3 - 3x^2y$, which again transforms according to a one-dimensional irrep ($A_{-+}$), is included in $\mathcal{M}$. Note that the addition of this extra polynomial does not require any new descendants beyond those already included. Hence, our multipole group can be summarized as

$$f(\mathbf{x}) \in \{(x^2 + y^2)^2 + \text{descendants}, y^3 - 3x^2y\}. \qquad (33)$$

Despite the highest-order polynomial being of degree four, we are able to find a unique third-order derivative that annihilates all polynomials: $\mathcal{D}_1 = \partial_x^3 - 3\partial_x\partial_y^2$. If we had not included $f(\mathbf{x}) = y^3 - 3x^2y$ in $\mathcal{M}$, we would additionally be able to introduce a second third-order derivative, $\mathcal{D}_2 = \partial_y^3 - 3\partial_x^2\partial_y$. We note in passing that the operator $\mathcal{D}_1$ also naturally appears in

the context of hydrodynamics in the presence of the point group $D_3$, since it may be written compactly as $\mathcal{D}_1 = \lambda_{ijk}\partial_i\partial_j\partial_k$, with $\lambda_{ijk}$ the third-order $D_3$-invariant tensor [52, 53].

At the level of continuum derivatives, the operator $\mathcal{D}_1$ on its own annihilates a much larger family of functions than those belonging to $\mathcal{M}$. This can be illustrated by acting on functions $e^{i\mathbf{k}\cdot\mathbf{x}}$, we observe that $\mathcal{D}_1$ acts multiplicatively as $\propto k_x(k_x + \sqrt{3}k_y)(k_x - \sqrt{3}k_y)$. That is, any linear combination of oscillatory functions that satisfy $k_x = 0$ or $k_x = \pm\sqrt{3}k_y$ (i.e., those that do not vary parallel to the triangular lattice directions, $\mathbf{e}_x$ or $\mathbf{e}_x \mp \sqrt{3}\mathbf{e}_y$, respectively) will be annihilated by $\mathcal{D}_1$. This leads to the same conservation laws as a *subsystem symmetry*: since $\mathcal{D}_1$ will annihilate $\delta(y)$ and analogous functions for the other two lattice directions, charge is conserved along every line that is parallel to each of the three lattice directions.

### 1. Lattice Hamiltonian

On the lattice, we can systematically construct discrete derivatives according to the procedure outlined in Sec. III A 1. We will restrict our attention to regions of the triangular lattice constructed by finding all sites contained within a circle of radius $\ell$. The origin will be arbitrary, but circles centered on sites will generate $D_6$-symmetric clusters while those centered on triangular plaquettes will generate $D_3$-symmetric clusters, etc. Further, we will examine the most highly constrained case corresponding to minimal on-site Hilbert space dimension: spin-1/2 degrees of freedom. Subject to these constraints, the solution with the smallest range is *unique* and consists of spin raising and lowering operators acting around a hexagon ($\ell = 1 + \epsilon$, with $\epsilon$ a positive infinitesimal):

$$D_1 \sim \quad \raisebox{-0.5em}{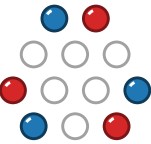} \quad , \tag{34}$$

where the red sites (say) correspond to spin raising operators, $\hat{S}^+$, and the blue sites to $\hat{S}^-$. We omit the integer labels utilized in the gates (30) and (32) since we are working with spin-1/2 degrees of freedom. Hence, if $(\mathbf{x}, 1), \ldots, (\mathbf{x}, 6)$ label the spins around a hexagon surrounding a lattice site $\mathbf{x}$ in a counter-clockwise direction [the colored sites in (34)], we have the "ring-exchange" gate

$$\hat{h}_\mathbf{x} = \hat{S}^+_{\mathbf{x},1}\hat{S}^-_{\mathbf{x},2}\hat{S}^+_{\mathbf{x},3}\hat{S}^-_{\mathbf{x},4}\hat{S}^+_{\mathbf{x},5}\hat{S}^-_{\mathbf{x},6}, \tag{35}$$

centered on sites, an interaction that is commonly found in the context of frustrated magnetism [54, 55]. The discrete derivative operator in Eq. (34), like the continuum derivative to which it coarse grains ($\mathcal{D}_1 = \partial_x^3 - 3\partial_x\partial_y^2$), conserves charge along lines. From (34), we observe that $\sum_{i\in\gamma} n_1(\boldsymbol{\delta}_i) = 0$ for all lines $\gamma$ that are parallel to the three triangular lattice directions. Therefore, for every closed line connecting sites of the triangular lattice that is parallel to one of the lattice directions, there exists a corresponding conserved charge. For a lattice of $L \times L$ primitive unit cells (see Fig. 1), there are $3L$ such conserved charges, although not all of these are independent. Hence,

the behavior of the discrete derivatives mirrors that of the continuum derivatives: in the discrete (continuous) case, the operator that annihilates all polynomial moments with smallest range (with fewest derivatives) conserves charge along lines parallel to lattice directions.

We may now enlarge the radius of the circle to look for discrete derivatives that annihilate the moments in (33) with larger range. Clearly, if the enlarged region includes multiple hexagons of the form (34), we can form valid discrete derivatives by taking linear combinations of the motif in (34) (as long as the resulting coefficients all satisfy the constraint $|n_\alpha(\mathbf{r})| \leq 1$). For instance, the operator

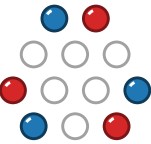

can be formed by taking a linear combination of solutions (34) on the central three sites forming an 'up' triangle, and will exhibit precisely the same conservation laws as (34). As a result, to find derivatives that do not preserve the constraint, we should restrict our search to operators that do not belong to the span of derivative operators of the form (34). The first such operator appears at the radius $\ell$ satisfying $2\ell = \sqrt{19} + \epsilon$:

$$\raisebox{-0.5em}{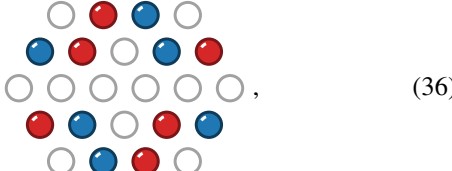} , \tag{36}$$

which ceases to conserve charge along lattice directions that are not parallel to $x$. Note that derivative operators belonging to the orbit of (36) under symmetry operations will also be valid solutions. Further note that we present the derivative operator with the smallest support by adding or subtracting operators of the form (34). The operator in (36) coarse grains to the *fourth*-order continuum derivative $\propto \partial_x(\partial_y^3 - 3\partial_x^2\partial_y)$. While the exact conservation law is broken at the lattice scale by gates such as (36), in the long-wavelength limit, the subsystem-symmetry-breaking gates are less relevant and lead to an infrared (IR) description in which subsystem symmetry is broken only by higher-order, dangerously irrelevant corrections.

While we originally discovered this emergent subsystem symmetry on the triangular lattice, it turns out that an analogous scenario can also obtain on the square lattice. If we work on the square lattice, with spin-1/2 degrees of freedom, and conserve $x^{2n} \pm y^{2n}$ or $\{x^{2n+1}, y^{2n+1}\}$, for even and odd orders, respectively, then these polynomials (and their descendants) are all annihilated by $\partial_x\partial_y$, which has subsystem symmetry along lines. The gate of minimal size that is compatible with the chosen multipolar symmetries is then just ring exchange around a square plaquette, the consequences of which were discussed in detail in Ref. [23]. Similar to the triangular lattice, there exist gates analogous to (36) that break the microscopic subsystem symmetry, but in principle we could always add

additional multipolar conservation laws to forbid the smaller subsystem-symmetry-breaking gates, thereby ensuring that the smallest gate that breaks subsystem symmetry is larger than any desired size.

### 2. *Haar-random circuits*

To test our prediction that subsystem symmetry emerges at sufficiently long length and time scales, we perform simulations of Haar-random circuits that preserve the relevant conserved quantities (33). In particular, we work with Haar-random gates that are large enough for subsystem symmetry to be broken at the microscopic level, i.e., such that gates analogous to (36) are included. For two-point correlation functions, we perform the Haar average *exactly*, which gives rise to an effective stochastic automaton dynamics that can be simulated efficiently [scaling as poly($L$)] for large systems (similar mappings exist for other quantities evaluated in Haar-random circuits, see, e.g., Refs. [56–59]). The mapping to automaton dynamics is outlined in detail in Appendix E. To summarize briefly, the Haar-random circuit is mapped to automaton dynamics that permits *all* symmetry-allowed transitions (with equal probability) within a local region defined by the gate applied at each step.

As shown in Ref. [33], the Ward identity for charge conservation on the triangular lattice in the presence of subsystem symmetry along lattice directions is

$$\partial_t \rho + \partial_1 \partial_2 \partial_3 J = 0 \,, \tag{37}$$

where the scalar current $J$ is related to the charge density $\rho$ via the constitutive relation $J = -\lambda \partial_1 \partial_2 \partial_3 \rho$ at leading order, and the derivatives $\partial_1$, $\partial_2$, and $\partial_3$ are directed along the three triangular lattice directions. Equation (37) gives rise to the highly anisotropic decay rate $\Gamma(\mathbf{k}) = \lambda k^6 \cos^2(3\theta)/16$, for density modulations of wave vector $\mathbf{k}$, with $(k, \theta)$ the polar coordinates of $\mathbf{k}$. That is, the decay rate $\Gamma(\mathbf{k})$ has flat directions along $\theta = \pi/6 + n\pi/3$ (arising directly from subsystem symmetry) and scales with the *sixth* power of $k$. The Ward identity (37) straightforwardly determines the correlation function of $\hat{S}_i^z$ through

$$\overline{\langle \hat{S}^z(\mathbf{r}; t) \hat{S}^z(\mathbf{0}; 0)\rangle} \simeq \frac{1}{3} S(S+1) \frac{1}{L^2} \sum_{\mathbf{k}} e^{i\mathbf{k}\cdot\mathbf{r}} e^{-\Gamma(\mathbf{k})t} \,, \tag{38}$$

where the sum is over wavevectors $\mathbf{k}$ compatible with the periodic boundary conditions, and the overline denotes an average over circuit geometries and of the gates over the circular unitary ensemble (CUE), i.e., a Haar average. The function (38) is contrasted with the output of the Haar-random circuit simulations in Fig. 9, which exhibit excellent agreement. There is just one free parameter: the phenomenological subdiffusion constant $\lambda$. In the thermodynamic limit, we find from (38) that $C_z(\mathbf{r}; t)$ decays slowly with distance as $|\mathbf{r}|^{-1/2}$ for $\mathbf{r}$ parallel to one of the three triangular lattice directions, leading to distinctive sharp features that are reproduced by the numerical simulations. The coincidence of the theoretical prediction (38) and the random quantum circuit result verifies that the late-time behavior of

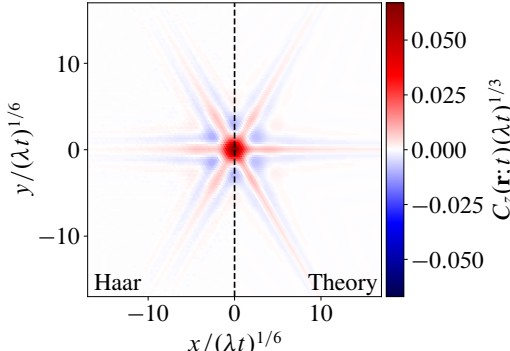

FIG. 9. Comparison between the autocorrelation function $C_z(\mathbf{r}; t) = \langle \hat{S}^z(\mathbf{r}; t) \hat{S}^z(\mathbf{0}; 0)\rangle$ obtained for a Haar-random circuit that conserves the multipole moments (33) (left) and the corresponding hydrodynamic prediction (38) (right). The correlation function is evaluated using an effective classical automaton evolution that allows us to reach large systems and times. The spatial profile is illustrated at a fixed time, $t = 5 \times 10^3$, for a system of linear size $L = 512$.

correlation functions under *generic* dynamics that conserves the moments in (33) is governed by an equation of motion that exhibits subsystem symmetry. That is, even though there is generically no subsystem symmetry at the microscopic level, it nevertheless emerges at late times and long wavelengths if we conserve the $O(1)$ list of multipole moments in Eq. (33).

### 3. *Discussion*

It is known that sub-maximal multipole groups can exhibit additional emergent conservation laws leading to unexpectedly slow dynamics controlled by dangerously irrelevant perturbations. The U(1) generalization of Haah's code is a central example of this [42]. Our formalism can be used to construct many more models exhibiting such exotic physics on arbitrary lattices. We have illustrated this by a particularly striking construction, where imposition of a finite number of multipole moments on the triangular lattice leads to a robust emergent subsystem symmetry broken only by gates acting on twenty four or more sites (a similar construction can retrospectively be performed on the square lattice). If we limit the range of the gates such that the subsystem symmetry becomes exact (albeit accidental), then we get the exotic lattice consequences discussed in Ref. [23], including, for example, finite thickness 'shields' that disconnect the system. If we allow for terms in the Hamiltonian with arbitrary range, then the subsystem symmetry is eventually broken, but the subsystem symmetry breaking perturbations are irrelevant such that long wavelength hydrodynamic behavior is still consistent with subsystem symmetry (unless we specifically examine relaxation of subsystem symmetry charges, in which case it will be controlled by dangerously irrelevant perturbations).

The constructions we present herein are interesting for multiple reasons. Firstly, of course, there is the exotic quantum dynamics and hydrodynamics discussed above. Next, Haah's code is a particularly interesting example of a fracton phase [60–

64], which continues to challenge several emerging paradigms (see, e.g., Refs. [65–67]). The U(1) generalization of Haah's code appears to inherit its exotic properties from a sub-maximal multipole group. We have provided a route to the construction of a multitude of sub-maximal multipole groups on arbitrary lattices. Going back from U(1) to $\mathbb{Z}_2$ might then provide a whole family of models analogous to Haah's code, opening a new chapter for the field. Finally, subsystem symmetry itself is of interest [44, 68–73], but is extremely unlikely to arise exactly in a microscopic Hamiltonian. We have provided a general construction for how subsystem symmetry may be emergently obtained from a finite number of conservation laws, which may also be useful for uncovering subsystem symmetries in real materials.

### D. Vector conserved quantities

In Sec. II C 3 we identified multipole groups for the breathing kagome lattice (Fig. 8) that did not require conservation of the $z$ component of total magnetization $\sum_i \hat{S}_i^z$ (i.e., did not require conservation of monopole charge). Here, we show that these multipolar conservation laws instead produce an emergent two-component *vector* conserved charge density, and that the smallest lattice derivatives additionally conserve moments of this vector density related to holomorphic complex functions.

Consider the following list of multipole moments, which, according to Table III, generate a valid multipole group on the breathing kagome lattice:

$$f_a(\mathbf{r}) \in \left\{ v_x, v_y, x_\delta v_x - y_\delta v_y, x_\delta v_y + y_\delta v_x \right\}, \quad (39)$$

where $x_\delta \equiv x - \delta_x$, with $\delta_x$ shorthand for the sublattice-dependent vector shift that appears in Eq. (17) (analogously for $y_\delta$). While the list of multipole moments in Eq. (39) looks somewhat unnatural, we are able to rewrite all position dependence of the polynomials in terms of the nearest Bravais lattice sites as

$$f_a(\mathbf{R}) \in \left\{ v_x, v_y, R_x v_x - R_y v_y, R_x v_y + R_y v_x \right\}, \quad (40)$$

where $\mathbf{R}$ corresponds to the position of the $C_3$ rotation center associated with the three sites (see Fig. 8), which are labeled by the index $a$. That is, when evaluating multipole moments, sites are weighted according to the position of the rotation center to which they are associated. This point is discussed in further detail and motivated physically in Sec. II C 3. Unlike the examples considered thus far, this multipole group does not require conservation of total charge, which would correspond to conserving the moment specified by $f_a = v_0 = (1, 1, 1)^T$. In what follows, it will be convenient to introduce the following set of unit basis vectors $\{\mathbf{e}^{(a)}\}$ for the triangular lattice

$$\mathbf{e}^{(1)} = \left( +\tfrac{\sqrt{3}}{2}, \tfrac{1}{2} \right), \ \ \mathbf{e}^{(2)} = \left( -\tfrac{\sqrt{3}}{2}, \tfrac{1}{2} \right), \ \ \mathbf{e}^{(3)} = (0, -1) \ . \quad (41)$$

Since the three vectors are related to one another by $C_3$ rotations, they satisfy $\sum_a \mathbf{e}^{(a)} = 0$. Note that this choice permits us to rewrite the conservation laws in terms of a *two*-component

quantity $F_k(\mathbf{R})$ via $f_a(\mathbf{R}) = e_k^{(a)} F_k(\mathbf{R})$. In terms of $F_k(\mathbf{R})$, the conservation laws are simplified to

$$F_k(\mathbf{R}) \in \left\{ \begin{pmatrix} 1 \\ 0 \end{pmatrix}, \begin{pmatrix} 0 \\ 1 \end{pmatrix}, \begin{pmatrix} R_x \\ R_y \end{pmatrix}, \begin{pmatrix} R_y \\ -R_x \end{pmatrix} \right\} \ . \quad (42)$$

Recall that the functions $f_a(\mathbf{R})$ define conserved charges via $\hat{\mathcal{Q}}[f] = \sum_{\mathbf{R},a} f_a(\mathbf{R}) \hat{S}_{\mathbf{R},a}^z$. Rewriting these conserved charges in terms of $F_k(\mathbf{R})$, we have

$$\hat{\mathcal{Q}}[F] = \sum_{\mathbf{R},k} F_k(\mathbf{R}) \hat{\rho}_k(\mathbf{R}), \quad (43)$$

where we have introduced the operators $\hat{\rho}_k(\mathbf{R}) \equiv e_k^{(a)} \hat{S}_{\mathbf{R},a}^z$, which live on the Bravais lattice site $\mathbf{R}$. The conservation laws in (39) and (40) therefore imply that, while the total scalar charge $\hat{\mathcal{Q}}[1] = \sum_{\mathbf{R},a} \hat{S}_{\mathbf{R},a}^z$ is *not* conserved, it is nevertheless possible to define an operator $\hat{\rho}_k(\mathbf{R})$ living on Bravais lattice sites with four conserved multipole moments:

$$\sum_{\mathbf{R}} \hat{\rho}_k(\mathbf{R}), \quad \sum_{\mathbf{R}} \mathbf{R} \cdot \hat{\boldsymbol{\rho}}(\mathbf{R}), \quad \sum_{\mathbf{R}} \mathbf{R} \times \hat{\boldsymbol{\rho}}(\mathbf{R}). \quad (44)$$

This follows from substituting (42) into (43). For convenience, we defined the two-dimensional cross product as the $z$ component of the conventional three-dimensional cross product. Note that $\hat{\rho}_k(\mathbf{R})$ indeed transforms as a vector under point group operations. Specifically, $C_3$ rotations, under which $\hat{\rho}_k \mapsto C_{k\ell} \hat{\rho}_\ell$, since three-fold rotations permute the spins associated to a given Bravais site ($C_{k\ell}$ is the rotation matrix for $\theta = 2\pi/3$), and the generating mirror $x \mapsto -x$, under which $\hat{\rho}_x \mapsto -\hat{\rho}_x$ and $\hat{\rho}_y \mapsto \hat{\rho}_y$.

As shown in Sec. III A, we can construct lattice Hamiltonians that satisfy these conservation laws by finding (discrete) derivatives that annihilate the list of conserved functions in Eq. (39) [or, equivalently, Eq. (42)]. While any second-order derivative will annihilate the polynomials in Eq. (42), there also exists a particular solution composed of *first*-order derivatives. Namely, the derivative operators $(\partial_x, -\partial_y)$ and $(\partial_y, \partial_x)$ will also annihilate all moments $F_k(\mathbf{R})$. However, these lowest-order derivatives additionally conserve a much larger family of functions beyond the finite list in Eq. (42). Any two-component function $g_k(\mathbf{r})$ that satisfies

$$-\partial_x g_x + \partial_y g_y = 0 \quad (45a)$$
$$\partial_y g_x + \partial_x g_y = 0 \quad (45b)$$

will be annihilated by the first-order derivative operators that we have identified. These equations are simply the Cauchy-Riemann equations whose solutions define the holomorphic functions. Hence, *any* holomorphic function will be annihilated by the pair of first-order derivatives $(\partial_x, -\partial_y)$ and $(\partial_y, \partial_x)$. In particular, the conserved first-order moments $\sum_{\mathbf{R}} \mathbf{R} \cdot \hat{\boldsymbol{\rho}}(\mathbf{R})$ and $\sum_{\mathbf{R}} \mathbf{R} \times \hat{\boldsymbol{\rho}}(\mathbf{R})$ appearing in Eq. (42) correspond to the holomorphic functions $g(z) = z$ and $g(z) = iz$, respectively, where the $x$ ($y$) component is recovered from the real (imaginary) part of the complex-valued function $g(z)$. More generally, the first-order derivatives will annihilate *all* functions spanned by $g(z) = z^n$ and $g(z) = iz^n$ for $n \in \mathbb{N}_0$. Such holomorphic

conserved charges also arise in the context of hydrodynamics in the presence of a triangular point group when the current tensor transforms in the vector representation of $D_3$ [52].

### 1. Lattice Hamiltonian

To put the theory on a lattice comprised of spin-1/2 degrees of freedom, we are tasked with finding discrete derivatives defined by a set of integer coefficients $n_\alpha(\delta)$ satisfying $|n_\alpha(\delta)| \leq 1$ that annihilate the list of functions in Eq. (39). Explicitly, for a cluster $C$ composed of Bravais sites $\mathbf{R}$ and basis sites $a$, we require that $(D_\alpha f)(\mathbf{x}) = \sum_{i \in C} n_{\alpha a_i}(\Delta_i) f_{a_i}(\mathbf{x} + \Delta_i) = 0$, where $\Delta_i$ is the displacement between the Bravais site associated with the site $i$ and the center of the cluster $\mathbf{x}$ (see Appendix D for further details). As before, we will consider clusters defined by finding all sites contained within a circle of radius $\ell$. Considering first a single 'down' triangle, we find the solution

$$D_0 \sim \raisebox{-0.5em}{\includegraphics[height=1.5em]{d0}} , \tag{46}$$

which adds or subtracts charge from the three sites associated to a given Bravais lattice site. The derivatives can also be expressed in terms of their action on discrete $F_i(\mathbf{R})$ via $(D_\alpha f)(\mathbf{x}) = \sum_{I \in C, a} e_k^{(a)} n_{\alpha a}(\Delta_I) F_k(\mathbf{x} + \Delta_I)$, where the index $I$ labels the Bravais lattice sites (we assume that the cluster $C$ encompasses all $a$ associated with each included Bravais site). We may therefore define the discrete derivative $(\tilde{D}_\alpha F)(\mathbf{x}) = \sum_{I \in C} \tilde{n}_{\alpha k}(\Delta_I) F_k(\mathbf{x} + \Delta_I)$, i.e., we define a set of coefficients $\tilde{n}_{\alpha k}(\Delta) \equiv \sum_a e_k^{(a)} n_{\alpha a}(\Delta)$ which are associated to Bravais lattice sites (the coefficients $\tilde{n}_\alpha$ may no longer be integer valued). Note that the gate $D_0$ in (46) leaves $\hat{\rho}_i$ unchanged since the basis vectors $\mathbf{e}^{(a)}$ sum to zero (equivalently, $\tilde{D}_0$ is the trivial derivative operator with $\tilde{n}_{\alpha i} = 0$). Next, we enlarge our search to include regions of that lattice that include up to three 'down' triangles. We find nontrivial solutions centered on both 'up' triangles and on hexagonal plaquettes

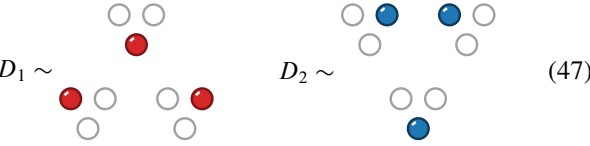

$$D_1 \sim \qquad\qquad D_2 \sim \tag{47}$$

The gates related to (47) by $C_3$ rotations are also valid solutions, which allows us to construct a Hamiltonian that is invariant under the space group. Up to these symmetry-equivalent solutions [and addition or subtraction of the solution in (46)], the gates in (47) are unique, although considering clusters of increasing size will eventually yield solutions that are linearly independent from (46) and (47). Upon coarse graining, the derivative operators $\tilde{D}_1$ and $\tilde{D}_2$ become precisely the operator

$(\partial_x, -\partial_y)$ identified previously, i.e.,

$$\tilde{D}_1 \sim \frac{1}{2}\left(\sqrt{3}\ \raisebox{-0.5em}{\includegraphics[height=1.5em]{d1a}}\ ,\ \raisebox{-0.5em}{\includegraphics[height=1.5em]{d1b}}\right) \rightarrow \frac{\sqrt{3}}{2}(\partial_x, -\partial_y) \tag{48a}$$

$$\tilde{D}_2 \sim \frac{1}{2}\left(\sqrt{3}\ \raisebox{-0.5em}{\includegraphics[height=1.5em]{d2a}}\ ,\ \raisebox{-0.5em}{\includegraphics[height=1.5em]{d2b}}\right) \rightarrow \frac{\sqrt{3}}{2}(\partial_x, -\partial_y) \tag{48b}$$

where the sites depicted now correspond to the triangular Bravais lattice formed by the centers of 'down' triangles. Their $C_3$-rotated variants coarse grain to a linear combination of $(\partial_x, -\partial_y)$ and $(\partial_y, \partial_x)$. Hence, as shown in Appendix D, the equations that define which functions are conserved by the Hamiltonian are precisely discrete versions of the Cauchy-Riemann equations (45), whose long-wavelength solutions will give rise to (approximate) holomorphic conserved charges. In a similar manner to Sec. IV C, even allowing for terms in the Hamiltonian with arbitrary range, the derivative operators that break holomorphic charge conservation are irrelevant. As a result, long-wavelength relaxation is determined by an equation of motion that preserves the holomorphic charges discussed herein.

### 2. Discussion

We have identified an exotic and (as far as we know) hitherto unknown possibility – a conserved multipole group that does *not* conserve monopole charge – and have identified a concrete realization on the breathing kagome lattice. We have pointed out that this multipole group conserves an emergent *vector* charge, plus all holomorphic functions of this vector charge. This constitutes a new and exotic class of problem, for which exploration of the thermodynamics and dynamics promises to be a particularly interesting challenge for future work. It also provides a 'proof of principle' of qualitatively new physics that is *only* accessible on non-hypercubic lattices, and thus can *only* be accessed through our formalism, or something analogous.

## V. CONCLUSIONS

We have presented an algorithmic procedure by means of which one may construct all consistent conserved multipole groups to any desired order on arbitrary crystal lattices, and may further construct the minimal continuum field theory and lattice Hamiltonian consistent with said conservation laws. The procedure for constructing consistent multipole groups is as follows: given a space group, compute the extended point group and sort polynomials in the spatial coordinates into irreps. Then, for each Wyckoff position, decompose the permutation action of the extended point group on the basis sites into irreps. Using Clebsch-Gordon coefficients, combine the above irreps into irreps of the full extended point group. Finally, compute the translation mixing. This method is dimension- and lattice-independent, and can thus be applied to arbitrary crystal lattices. As such, it provides a complete *in principle* classification of the multipolar problem on arbitrary lattices. The procedure is labor intensive, so we have only carried it out for a representative set of

lattices in two dimensions (all two dimensional Bravais lattices, plus kagome and breathing kagome). However, extension to, e.g., all 230 space groups in three spatial dimensions would be straightforward, albeit tedious. An explicit classification for all known crystal structures would be a worthwhile challenge for future work.

We have explored some interesting physical consequences using our construction. As a warmup, we have identified the minimal set of symmetries required to get localization from strong fragmentation on the square and triangular lattices respectively (using a method that could be generalized to any lattice). We have also identified two phenomena that do not appear to have been appreciated before. One involves an emergent *subsystem* symmetry arising from imposition of a finite number of multipolar conservation laws. The second new phenomenon has no known analog on hypercubic lattices whatsoever – it turns out that on the breathing kagome lattice, one can define a consistent multipole group that does *not* include monopole. Thus, one can write down consistent translation invariant Hamiltonians that conserve certain multipole moments of charge, but do not conserve total charge itself! Nevertheless, these models do conserve an emergent 'vector' charge, as well as all holomorphic functions thereof. This constitutes a striking and novel scenario, deserving of more detailed exploration in future work.

Our formalism opens up new directions for multiple lines of research. At the most basic level, it provides a guide to the construction of new fracton models on non-hypercubic lattices.[5] Particularly interesting, it allows us a way to identify consistent *sub-maximal* multipole groups. Given that the sub-maximal group appears to be the 'secret ingredient' that endows Haah's [U(1)] code with its uniquely exotic properties, it offers a route to the construction of a whole family of models analogous to Haah's code, on lattices other than cubic. Haah's code is the least well understood fracton model, challenging many paradigms [65–67], and construction of a family of models analogous to Haah's code might open up new directions for research into fracton phases. Given that Haah's code has interesting properties as a quantum memory [74, 75], such models may also be useful for quantum information. On another front, subsystem symmetries are of considerable current interest [44, 68–70], but given that subsystem symmetries involve infinitely many conservation laws, it is not clear how such symmetries could arise in nature. We have provided constructions through which imposition of a finite number of multipolar conservation laws can lead to the emergence of subsystem symmetry, which could help guide the search for realizations of subsystem symmetries in nature. (It should be noted, however, that multipolar symmetries other than dipolar are already challenging to realize, as discussed in Ref. [12]). On a third front, our explorations have led us to discover new phenomena that have *no* known analog on (hyper)cubic lattices – for instance the possibility of having systems that do not conserve charge, but do conserve certain multipolar

moments of charge. This opens a new direction for the study of fracton phases and associated phenomena on general lattices, and demonstrates that there is qualitatively new physics to be found. And finally, our work can guide the search for fracton physics and associated phenomena in real materials – many of which are not based on square or cubic lattices. Note that crisp experimental diagnostics for fracton phases have been identified in Refs. [76–78].

Finally, we have thus far limited ourselves to systems where all symmetries of the Hamiltonian are preserved. It would be very interesting to examine spontaneous symmetry breaking of crystalline multipole groups. Either the space group part or the polynomial shift part of the symmetry could be broken, and the interplay between discrete spatial symmetries and continuous polynomial shift symmetries could lead to interesting effects. One could even consider exotic possibilities like breaking a spatial symmetry and a polynomial shift symmetry but preserving the product. An exploration of such exotic symmetry breaking phenomena is an(other) interesting challenge for future work.

**ACKNOWLEDGMENTS**

OH is grateful to Marvin Qi for useful discussions relating to the holormorphic conserved charges. We also thank Paul Romatschke and Ted Nzuonkwelle for providing access to the Eridanus cluster at CU Boulder. This work was supported by the U.S. Department of Energy, Office of Science, Basic Energy Sciences, under Award #DE-SC0021346.

**Appendix A: Group theory**

Since our results rely heavily on sorting polynomials into irreps of finite groups, such as $D_M$, we briefly list their irreps and summarize the notation that we use to denote them.

### A.1. Irreducible representations of $D_M$

Let $r$ denote reflection through a fixed symmetry axis of an $M$-gon, and let $C$ denote a rotation through an angle $\theta_M = 2\pi/M$ about the center of the $M$-gon. The group $D_M$ can then be written as

$$D_M = \left\langle C, r \,\middle|\, C^M = r^2 = \mathbb{I}, rCr = C^{-1} \right\rangle. \tag{A.1}$$

The irreps of $D_M$ depend on whether $M$ is even or odd. For even $M$, there are four one-dimensional irreps and $M/2 - 1$ two-dimensional irreps. We use the following notation for the four one-dimensional irreps:

$$A_{\sigma\nu}(C) = \sigma, \ A_{\sigma\nu}(r) = \nu, \tag{A.2}$$

with $\sigma, \nu \in \{1, -1\}$. The two-dimensional irreps are indexed by an integer $k = 1, \ldots, M/2 - 1$, and are denoted

$$E_k(C) = \begin{pmatrix} \cos k\theta_M & -\sin k\theta_M \\ \sin k\theta_M & \cos k\theta_M \end{pmatrix}, \ E_k(r) = \begin{pmatrix} -1 & 0 \\ 0 & 1 \end{pmatrix}. \tag{A.3}$$

---

[5] Strictly, construction of a fracton model requires identification of a consistent multipole group followed by *gauging* of the polynomial shift part thereof.

When $M$ is instead odd, the two one-dimensional irreps $A_{-+}$ and $A_{--}$ are removed, leaving the two irreps $A_{++}$ and $A_{+-}$.

## A.2. Sorting polynomials into irreps

There are two algorithmic ways to sort polynomial shift symmetries into irreps of the extended point group. The first method is a simple one to perform on a computer. Fix a Wyckoff position of multiplicity $w$. By construction, each extended point group operation maps a collection of $w$ homogeneous polynomials of degree $n$ to another collection of homogeneous polynomials of degree $n$, so we may restrict our attention to the case where all basis sites transform via a homogeneous polynomial of degree $n$. The set of all such transformations is spanned by the transformations $f_i(\mathbf{r}) = \delta_{i,i_0} x^m y^{n-m}$ for all $i_0 = 1, 2, \ldots, w$ and $m = 0, 1, \ldots, n$. Viewing these transformations as a basis $|i_0, m\rangle$ for the set of polynomial shift symmetries under consideration, it is easy to compute directly how extended point group operations act on these transformations. For example, for $n = 2$ on the honeycomb lattice, rotations take

$$C\,|1, 2\rangle = C \begin{pmatrix} x^2 \\ 0 \end{pmatrix} = \begin{pmatrix} 0 \\ \left(\frac{1}{2}x + \frac{\sqrt{3}}{2}y\right)^2 \end{pmatrix}$$

$$= \frac{1}{4}\,|2, 2\rangle + \frac{\sqrt{3}}{2}\,|2, 1\rangle + \frac{3}{4}\,|2, 0\rangle \quad \text{(A.4)}$$

This gives us a matrix representation of each extended point group operation. Given such a matrix representation $M(g)$ for $g \in G$, where $G$ is the group in question, and given the characters $\chi_\ell(g)$ where $\ell$ labels a representation of $G$, one can construct the projector $P_\ell$ onto the representation $\ell$ via the formula

$$P_\ell = \frac{1}{|G|} \sum_{g \in G} \chi_\ell(g) M(g). \quad \text{(A.5)}$$

The eigenvectors of this projector with eigenvalue 1 are a basis for the polynomials which transform under the given representation $\ell$. One can then convert this into any convenient basis.

The second method is analytical and uses the Clebsch-Gordon coefficients of the extended point group. First, one can decompose the polynomials $\{x, y\}$ into irreps of the pure coordinate transformations $\mathbf{r} \to \mathbf{s}\mathbf{r}$. All higher-order polynomials are formed as (tensor) products of some of these irreps with themselves; using Clebsch-Gordon coefficients, one can decompose those tensor products into irreps of these pure coordinate transformations. For example, under the extended point

group $\mathsf{D}_4$, $\{x, y\}$ forms the two-dimensional representation $E_1$. Hence quadratic polynomials all appear in the tensor product $E_1 \otimes E_1 = A_{++} \oplus A_{+-} \oplus A_{-+} \oplus A_{--}$. Using the Clebsch-Gordon coefficients, we find that the $A_{++}$ representation is $x^2 + y^2$, the $A_{--}$ representation is $2xy$, the $A_{-+}$ representation is $x^2 - y^2$, and the $A_{+-}$ representation is 0 (and thus not present). This procedure can then be iterated; cubic polynomials are formed from (tensor) products of the linear polynomials and quadratic polynomials, and so on. A similar, related approach is to restrict polynomial representations of $\mathsf{O}(2)$ to representations of $\mathsf{D}_M$ via "branching rules" [79, 80].

This is the complete classification procedure if the lattice is a Bravais lattice. In the presence of a basis, we notice that the permutation action of the extended point group on the fields, and in particular on each Wyckoff position, also produces a (not necessarily irreducible) representation of the extended point group. This representation is also the representation formed by constant polynomial shift symmetries (since the coordinate transformation does nothing to constant shift symmetries) and can be decomposed straightforwardly into irreps, potentially using the projector method as above. The overall action of the extended point group on the fields is the tensor product of this permutation action with the action of pure coordinate transformations. Hence the decomposition into irreps consists of choosing a permutation irrep (i.e., constant polynomial shift symmetry), tensoring with a coordinate transformation irrep (i.e., a polynomial), and using Clebsch-Gordon coefficients to decompose the tensor product into irreps. For example, on the honeycomb lattice the permutation action is represented as the $A_{++} \oplus A_{--}$ representation of $\mathsf{D}_6$, where the $A_{++}$ represents sublattice-even shift symmetries and $A_{--}$ represents sublattice-odd symmetries. Hence, each irrep of the extended point group is just the tensor product of the irreps formed by pure polynomials (which are identical to those of the triangular lattice) with one of these permutation irreps. Since the permutation irreps are all 1D, the Clebsch-Gordon coefficients are very simple.

## A.3. Clebsch-Gordon coefficients for $\mathsf{D}_M$

For convenience, we list the Clebsch-Gordon coefficients for $\mathsf{D}_M$ in the reflection eigenbasis. One derivation is given in Ref. [81] in the rotation eigenbasis. We make a notation change for this appendix; instead of referring to the one-dimensional irreps of $\mathsf{D}_M$ as $A_{\sigma,\nu}$, we will refer to them as $A_{\mu,\nu}$ for $\mu = 0, M/2$, where $\mu = 0$ corresponds to $\sigma = +1$ and $\mu = M/2$ corresponds to $\sigma = -1$. This notation makes several of the Clebsch-Gordon coefficients significantly simpler.

We first give the rules for which representations appear in the tensor product:

$$A_{\mu,\nu} \otimes A_{\mu',\nu'} = A_{\mu+\mu',\nu\nu'} \tag{A.6}$$

$$A_{\mu,\nu} \otimes E_{\mu'} = E_{\mu+\mu'} = E_{-(\mu+\mu')} \tag{A.7}$$

$$E_\mu \otimes E_{\mu'} = \begin{cases} A_{0,+} \oplus A_{0,-} \oplus E_{2\mu} & \mu = \mu' \neq \frac{M}{4}, \\ A_{0,+} \oplus A_{0,-} \oplus A_{M/2,+} \oplus A_{M/2,-} & \mu = \mu' = \frac{M}{4} \text{ if } M \text{ even,} \\ A_{M/2,+} \oplus A_{M/2,-} \oplus E_{\mu-\mu'} & \mu + \mu' = \frac{M}{2} \text{ and } \mu \neq \mu', \\ E_{\mu+\mu'} \oplus E_{\mu-\mu'} & \text{else.} \end{cases} \tag{A.8}$$

The Clebsch-Gordon coefficients themselves are only non-trivial when at least one representation involved is 2D. We label the basis for a 2D irrep $E_\mu$ as $|\mu; \pm\rangle$ where the generating mirror $r$ acts as

$$r |\mu; \pm\rangle = \pm |\mu; \pm\rangle . \tag{A.9}$$

Suppose the 1D irrep $A_{\mu,\nu}$ acts on the 1D vector space spanned by $|A_{\mu,\nu}\rangle$. Then the coefficients for $A_{\mu,\nu} \otimes E_{\mu'}$ are,

$$|\mu + \mu'; +\rangle = |A_{\mu,\nu}\rangle \otimes |\mu', \nu\rangle \tag{A.10a}$$

$$|\mu + \mu'; -\rangle = |A_{\mu,\nu}\rangle \otimes |\mu', -\nu\rangle \tag{A.10b}$$

Before giving the coefficients for $E_\mu \otimes E_{\mu'}$, we recall that $E_\mu = E_{-\mu}$; in the formulas that follow, we take the convention $\mu' > 0$ and $\mu$ takes whatever sign is appropriate to produce the value of $\mu + \mu'$ in question. When considering $E_\mu \otimes E_{\mu'}$, one generally must consider both $\mu$ and $-\mu$ to obtain all possible irreps in the product. For $E_\mu \otimes E_{\mu'}$, we have

$$|A_{\mu+\mu',\nu}\rangle = \frac{1}{\sqrt{2}} \left( |\mu; +\rangle \otimes |\mu'; \nu\rangle - \nu \, \mathrm{sgn}(\mu) \, |\mu; -\rangle \otimes |\mu'; -\nu\rangle \right) \tag{A.11}$$

when $\mu$ and $\mu'$ satisfy $\mu + \mu' \in \{0, \frac{M}{2}\}$, and

$$|\mu + \mu'; \nu\rangle = \frac{1}{\sqrt{2}} \left( |\mu; +\rangle \otimes |\mu'; \nu\rangle - \nu \, \mathrm{sgn}(\mu) \, |\mu; -\rangle \otimes |\mu'; \nu\rangle \right) \tag{A.12}$$

## Appendix B: Vector charge theory

In the main text, we assume that the fields in question transform as scalars under space group operations. This need not be the case; here, we show how the multipole group classification problem is modified if the field is not a scalar. In particular, this is necessary to produce a multipole symmetry group that, when gauged, produces the (2+1)D symmetric tensor vector charge theory [1, 82, 83] with Gauss' Law

$$\partial_i E_{ij} = \rho_j . \tag{B.1}$$

Suppose that we have a square lattice with two sites per unit cell, one on each bond of the square lattice. We call the fields on the $x/y$-directed bonds $\phi_{x/y}(\mathbf{r})$. Take the space group to be $p4m$, with point group $D_4$, and we assume that the fields

transform under point group operations as

$$C : \begin{pmatrix} \phi_x(\mathbf{r}) \\ \phi_y(\mathbf{r}) \end{pmatrix} \to \begin{pmatrix} \phi_y(M_C\mathbf{r}) \\ -\phi_x(M_C\mathbf{r}) \end{pmatrix} \tag{B.2a}$$

$$r : \begin{pmatrix} \phi_x(\mathbf{r}) \\ \phi_y(\mathbf{r}) \end{pmatrix} \to \begin{pmatrix} -\phi_x(M_r\mathbf{r}) \\ \phi_y(M_r\mathbf{r}) \end{pmatrix} \tag{B.2b}$$

where $C$ is a four-fold rotation, and the generating mirror $r$ sends $x \leftrightarrow -x$. The minus signs on the field are the new, crucial ingredient; we have assumed that $\phi_i$ transforms like a vector under $D_4$. The constant polynomial shift symmetries $(1,0)^T$ and $(0,1)^T$ now transform as the 2D irrep $E_1$ of $D_4$; the only way to conserve total charge while staying consistent with the space group is to conserve a vector-valued charge, where the components are the $\phi_x$ charge and $\phi_y$ charge. This is indeed a conserved charge of the vector charge theory, as expected, and it is quite different from the case where the fields are simply permuted under point group operations. At degree one, the polynomial shift symmetries in question are all one dimensional; labeling representations of $D_4$ as in Appendix A, they are $(x, -y)^T$ $(A_{++})$, $(x, y)^T$ $(A_{-+})$, $(y, x)^T$ $(A_{+-})$, and $(y, -x)^T$ $(A_{--})$. Choosing the multipole group to be generated by $(y, -x)^T$, $(1, 0)^T$, and $(0, 1)^T$ (the first set of polynomials generates the latter two via translations) produces the usual vector charge theory, which conserves the corresponding charges $\int d^2\mathbf{r} \, (\mathbf{r} \times \boldsymbol{\rho})_z$ and $\int d^2\mathbf{r} \, \boldsymbol{\rho}$.

## Appendix C: Discrete vs continuous translations

Given some vector of polynomials $f_i(\mathbf{r})$ in the multipole group, consider performing any discrete translation that could appear as part of a space group operation:

$$f_i(\mathbf{r}) \to f_i(\mathbf{r} + \ell \mathbf{a}_i) \tag{C.1}$$

where $\mathbf{a}_i$ is a lattice translation (that may in general depend on $i$) and $\ell$ is any integer that represents the number of times the discrete translation in question is being performed. We require that $f_i(\mathbf{r} + \ell \mathbf{a}_i)$ is also in the multipole group for every $\ell$ in order to have a closed multipole group. Listing out these polynomials is in general very tedious, as they will generally contain monomials of all degrees less than the degree of $f_i(\mathbf{r})$ (call this degree $n$). We claim that the collection of polynomials spanned by $f_i(\mathbf{r} + \ell \mathbf{a}_i)$ for all $\ell$ is the same as the collection of polynomials spanned by the procedure discussed in the main

text, namely where we formally take $\ell$ infinitesimal, generate a collection of polynomials of degree $n-1$, and then repeat the process for the newly generated polynomials.

Without loss of generality, let $\mathbf{a}_i$ be oriented along the $x$-axis; for a general direction, all partial derivatives may be replaced by directional derivatives. The power series expansion of a polynomial $f$ of finite degree yields

$$f(\mathbf{r} + \ell\mathbf{a}_i) = \sum_{j=0}^{n} \frac{(\ell|a|)^j}{j!} \frac{\partial^j}{\partial x^j} f(\mathbf{r}) . \qquad (C.2)$$

The highest term $n$ is the maximum power of $x$ appearing in $f$, which is, of course, bounded from above by the degree of $f$. Observe that $\partial^j f/\partial x^j$ always contains a term involving $x^{n-j} y^k$ (for some fixed power $k$) but never any terms involving a larger power of $x$. Therefore, the $n+1$ polynomials $\partial^j f/\partial x^j$ are linearly independent as elements of the vector space of polynomials over $\mathbb{R}$. We claim that this set of $n+1$ polynomials spans the same subspace as the set of $f(\mathbf{r} + \ell\mathbf{a}_i)$ for all integer $\ell$. Equation (C.2) immediately shows that the span of the latter set is contained in the span of the former. To see the other way around, consider the $n+1$ polynomials given by $\ell = 1, 2, \ldots, n+1$. Writing these polynomials in the basis given by $\partial^j f/\partial x^j$, we obtain a set of vectors

$$\begin{pmatrix} 1 & 1^1 & 1^2 & \cdots & 1^n \\ 1 & 2^1 & 2^2 & \cdots & 2^n \\ \vdots & \vdots & \vdots & \ddots & \vdots \\ 1 & (n+1)^1 & (n+1)^2 & \cdots & (n+1)^n \end{pmatrix} . \qquad (C.3)$$

This is a Vandermonde matrix with nonzero determinant since no two of its rows are equal. Therefore, it is invertible; its inverse is therefore a basis transformation that expresses the $\partial^k f/\partial x^j$ as a linear combination of the $f(\mathbf{r} + \ell\mathbf{a}_i)$ with our chosen values of $\ell$, so the span of the derivatives is contained in the span of the translates. In particular, $\partial f/\partial x$ is in the span of the translates. By symmetry, $\partial f/\partial y$ is as well.

We summarize the above argument as follows: given a lattice translation $\mathbf{a}_i$, the directional derivative $(\mathbf{a}_i \cdot \nabla)^j f_i$ for all $j$ span the same space of polynomials as the translates $f_i(\mathbf{r} + \ell\mathbf{a}_i)$ for all $\ell$. It remains to show that given another lattice translation $\mathbf{a}'_i$, the mixed derivatives $(\mathbf{a}_i \cdot \nabla)^j (\mathbf{a}'_i \cdot \nabla)^k f_i$ spans the same space of polynomials as $f_i(\mathbf{r} + \ell\mathbf{a}_i + m\mathbf{a}'_i)$. We can simply repeat the above argument, but applied to $(\mathbf{a}'_i \cdot \nabla)^k f_i$. Then we can conclude that $(\mathbf{a}_i \cdot \nabla)^j (\mathbf{a}'_i \cdot \nabla)^k f_i$ spans the same polynomials as $(\mathbf{a}'_i \cdot \nabla)^k f_i(\mathbf{r} + \ell\mathbf{a}_i)$. Applying the same argument again, we see that the latter spans the same polynomials as $f_i(\mathbf{r} + \ell\mathbf{a}_i + m\mathbf{a}'_i)$, as desired.

## Appendix D: From discrete derivatives to spin Hamiltonians

### D.1. Bravais lattice

Consider a given discrete derivative $D_\alpha$ on a Bravais lattice defined by its action on discrete functions $(D_\alpha f)(\mathbf{x}) = \sum_{i \in C} n_\alpha(\boldsymbol{\delta}_i) f(\mathbf{x} + \boldsymbol{\delta}_i)$. The real coefficients $n_\alpha(\boldsymbol{\delta}_i)$ are asso-

ciated with site $i$, which is displaced by $\boldsymbol{\delta}_i$ from the center of the cluster of sites $C$ (which we take to be $\sum_{i \in C} \boldsymbol{\delta}_i$). Note that $\mathbf{x}$, the displacement of the center of the cluster, might not be centered on sites of the Bravais lattice (e.g., $\mathbf{x}$ may correspond to the plaquettes or the bonds of the lattice). The positions $\mathbf{x} + \boldsymbol{\delta}_i$ always correspond to Bravais lattice sites, however. Suppose that the coefficients $n_\alpha(\boldsymbol{\delta})$ are all integer valued (perhaps by removing a common factor). If the integer-valued coefficients satisfy $|n_\alpha(\boldsymbol{\delta})| \leq 2S$, we can then define a Hamiltonian using the discrete derivative $D_\alpha$ acting on spin-$S$ degrees of freedom:

$$\hat{H} = \sum_{\mathbf{x}} \hat{h}_{\mathbf{x}\alpha} + \hat{h}^\dagger_{\mathbf{x}\alpha} , \qquad (D.1)$$

where we have defined the local "gate" $\hat{h}_{\mathbf{x}\alpha}$ associated to the discrete derivative $D_\alpha$ acting on the sites belonging to the cluster $C$ centered on $\mathbf{x}$

$$\hat{h}_{\mathbf{x}\alpha} = \prod_{i \in C} \left( \hat{S}^{\text{sgn}(n_\alpha(\boldsymbol{\delta}_i))}_{\mathbf{x}+\boldsymbol{\delta}_i} \right)^{|n_\alpha(\boldsymbol{\delta}_i)|} . \qquad (D.2)$$

The sign of the coefficients, $\text{sgn}(n_\alpha(\boldsymbol{\delta})) \in \{+, 0, -\}$ (where $\text{sgn}\, 0 = 0$), determine whether the site $\boldsymbol{\delta}$ in the cluster is associated to a spin raising or lowering operator. Turning the problem around, given spin-$S$ degrees of freedom, the restriction $|n_\alpha(\boldsymbol{\delta})| \leq 2S$ can impose strong constraints on the discrete derivatives that form valid Hamiltonians. In the most highly constrained case – spin-$1/2$ degrees of freedom – permitted derivative operators must satisfy $n_\alpha \in \{-1, 0, 1\}$ only. The utility of the construction in (D.1) is that the Hamiltonian conserves the moments of "charge" defined by functions $f$ that are annihilated by the discrete derivative $D_\alpha$. Explicitly, consider the putatively conserved operator $\hat{\mathcal{Q}}[f]$, parameterized by the function $f(\mathbf{r})$

$$\hat{\mathcal{Q}}[f] = \sum_{\mathbf{r}} f(\mathbf{r}) \hat{S}^z_{\mathbf{r}} , \qquad (D.3)$$

which corresponds to the $f(\mathbf{r})$ moment of the local "charge density" $\hat{S}^z_{\mathbf{r}}$, with total charge $\hat{\mathcal{Q}}[1] = \sum_{\mathbf{r}} \hat{S}^z_{\mathbf{r}}$ being recovered for the unit function $f(\mathbf{r}) = 1$. Making use of the commutation relations $[\hat{S}^z_i, (\hat{S}^{\text{sgn}\, n}_j)^{|n|}] = n\delta_{ij} (\hat{S}^{\text{sgn}\, n}_i)^{|n|}$ ($n \in \mathbb{Z}$) for spin-$S$ degrees of freedom, we find that

$$[\hat{\mathcal{Q}}[f], \hat{H}] = \sum_{\mathbf{x}} \left( \sum_{i \in C} n_\alpha(\boldsymbol{\delta}_i) f(\mathbf{x} + \boldsymbol{\delta}_i) \right) (\hat{h}_{\mathbf{x}} - \hat{h}^\dagger_{\mathbf{x}}) \quad (D.4a)$$

$$\equiv \sum_{\mathbf{x}} (D_\alpha f)(\mathbf{x}) (\hat{h}_{\mathbf{x}} - \hat{h}^\dagger_{\mathbf{x}}) . \qquad (D.4b)$$

That is, the commutator between $\hat{\mathcal{Q}}[f]$ and the Hamiltonian in (D.1) composed of local gates effects a discrete derivative on $f$. Hence, any function $f$ satisfying $D_\alpha f = 0$ defines a corresponding conserved charge $\partial_t \hat{\mathcal{Q}}[f] = 0$. In particular, if the coefficients $n_\alpha(\boldsymbol{\delta})$ satisfy $\sum_{i \in C} n_\alpha(\boldsymbol{\delta}_i) = 0$, then $D_\alpha$ will annihilate the unit function $f(\mathbf{r}) = 1$ and total charge $\hat{\mathcal{Q}}[1]$ will be conserved by dynamics generated by (D.1). Note that the arguments can also be applied in reverse: Given a Hamiltonian of the form (D.1), one can identify a family of

conserved charges $\hat{\mathcal{Q}}[f]$ by solving the equations $D_\alpha f = 0$.

## D.2. Introducing a basis

Now consider introducing a $q$-spin basis at each site of the Bravais lattice labeled by an index $a = 1, \ldots, q$. The action of a discrete derivative is still $(D_\alpha f)(\mathbf{x}) = \sum_{i \in C} n_\alpha(\boldsymbol{\delta}_i) f(\mathbf{x} + \boldsymbol{\delta}_i)$, where $\boldsymbol{\delta}_i$ is now the displacement of the *basis* site $i$ from the center of the cluster $C$. However, we can alternatively index the derivative coefficients and the function $f$ according to Bravais sites and a basis index. In this case the discrete derivative may be written

$$(D_\alpha f)(\mathbf{x}) = \sum_{i \in C} n_{\alpha a_i}(\boldsymbol{\Delta}_{I(i)}) f_{a_i}(\mathbf{x} + \boldsymbol{\Delta}_{I(i)}), \qquad \text{(D.5)}$$

where $\mathbf{x} + \boldsymbol{\Delta}_{I(i)}$ corresponds to the position of a Bravais site and $I(i)$ labels the Bravais lattice site associated to site $i$. That is, the vectors $\boldsymbol{\Delta}_I$ are displacements of the Bravais sites from the center of the cluster (now defined as $\sum_{I \in C} \boldsymbol{\Delta}_I$, which is identical to $q^{-1} \sum_{i \in C} \boldsymbol{\Delta}_{I(i)}$ if $C$ contains all basis sites associated with each Bravais site). Note that this is merely a reparametrization of the discrete function. Specifically, we are not assuming that the function depends only on the position of the Bravais site. For example, in the labeling scheme employed in (D.5), the function $f(\mathbf{r}) = \mathbf{r}^2$ would become $f_a(\mathbf{R}) = (\mathbf{R} + \boldsymbol{\delta}_a)^2$, where $\boldsymbol{\delta}_a$ is the vector that connects Bravais site $\mathbf{R}$ to basis site $\mathbf{r}_a$. In this language, the putatively conserved quantities $\hat{\mathcal{Q}}[f]$ now depend on a function $f_a(\mathbf{R})$ that itself depends on both the Bravais lattice site $\mathbf{R}$ and the index $a$:

$$\hat{\mathcal{Q}}[f] = \sum_{\mathbf{R},a} f_a(\mathbf{R}) \hat{S}^z_{\mathbf{R},a}. \qquad \text{(D.6)}$$

Again, we stress that whether $f(\mathbf{r})$ is a function of $\mathbf{r}$ or $\mathbf{R}(\mathbf{r})$ is a choice that is determined by the physics of the problem, but both cases are handled by the notation in (D.6). Suppose that the Hamiltonian now comprises multiple gates labeled by the integer $\alpha$

$$\hat{H} = \sum_{\mathbf{x},\alpha} g_\alpha (\hat{h}_{\mathbf{x}\alpha} + \hat{h}^\dagger_{\mathbf{x}\alpha}), \qquad \text{(D.7)}$$

with coupling constants $g_\alpha$. Repeating the calculation that led to (D.4a), we find that the time evolution of the charge $\hat{\mathcal{Q}}[f]$ is determined by

$$\partial_t \hat{\mathcal{Q}} \propto \sum_{\mathbf{x},\alpha} g_\alpha \left( \sum_{i \in C} n_{\alpha a_i}(\boldsymbol{\Delta}_{I(i)}) f_{a_i}(\mathbf{x} + \boldsymbol{\Delta}_{I(i)}) \right) \left( \hat{h}_{\mathbf{x}\alpha} - \hat{h}^\dagger_{\mathbf{x}\alpha} \right) \qquad \text{(D.8)}$$

which implies that $\hat{\mathcal{Q}}[f]$ is a conserved quantity under dynamics generated by (D.7) if *all* discrete derivatives annihilate the function $f_a(\mathbf{R})$.

## D.3. Additional symmetries

Note that the Hamiltonians in Eqs. (D.1) and (D.7) possess additional symmetries, as pointed out in, e.g., Refs. [13, 29]. For instance, the parity operator $\hat{\Pi}_x \equiv \prod_i e^{i\pi \hat{S}^x_i}$ commutes with the Hamiltonians (D.1) and (D.7) since $\hat{\Pi}_x$ has the effect of interchanging $\hat{S}^+_i$ with $\hat{S}^-_i$, i.e., $\hat{\Pi}_x \hat{S}^\pm_i \hat{\Pi}_x = \hat{S}^\mp_i$, therefore interchanging the gates $\hat{h}_{\mathbf{x}\alpha} \leftrightarrow \hat{h}^\dagger_{\mathbf{x}\alpha}$. We may, however, add any perturbation that is diagonal in $\hat{S}^z_i$ basis to the Hamiltonian while maintaining the conserved operators (D.3) and (D.6), since this will not affect the commutators in Eqs. (D.4a) and (D.8). Because the discrete symmetry $\hat{\Pi}_x$ anticommutes with $\hat{S}^z_i$, it only remains a conserved quantity if the perturbation consists of an even number of $\hat{S}^z_i$ operators. The anticommutation of $\hat{\Pi}_x$ and $\hat{S}^z_i$ also implies that $\hat{\Pi}_x$ and the multipole moments $\hat{\mathcal{Q}}[f]$ anticommute. Hence, if $\hat{\Pi}_x$ does commute with the Hamiltonian, the sectors with quantum numbers $\{Q[f]\}$ and $\{-Q[f]\}$, for all $f$ that are conserved by $\hat{H}$, will have precisely the same spectrum.

## D.4. More general Hamiltonians

As noted in the previous subsection, the Hamiltonians (D.1) and (D.7) will conserve the same family of charges $\hat{\mathcal{Q}}[f]$ if they are subjected to arbitrary perturbations that are diagonal in the $\hat{S}^z_i$ basis. In fact, the statement is more general: even the gates $\hat{h}_{\mathbf{x}\alpha}$ themselves can be 'decorated' by operator insertions that commute with $\hat{S}^z_i$. To illustrate this, consider a one-dimensional lattice (with no basis) that hosts a theory conserving only total charge $\sum_i \hat{S}^z_i$. The smallest gate of the form (D.2) that can be written down is simply $\hat{h}_{i,1} = \hat{S}^-_i \hat{S}^+_{i+1}$, which hops 'charge' one unit to the right. The next smallest operator of the form (D.2) is $\hat{h}_{i,2} = \hat{S}^-_{i-1} \hat{S}^+_{i+1}$, which hops charge two units to the right. However, taking the product of two adjacent gates $\hat{h}_{i-1,1} \hat{h}_{i,1} = \hat{S}^-_{i-1} \hat{S}^+_i \hat{S}^-_i \hat{S}^+_{i+1}$ differs from $\hat{h}_{i,2}$ by the operator $\hat{S}^+_i \hat{S}^-_i$ on the central site (physically, a hop of two units to the right cannot be exactly decomposed into two sequential hops, since charge on the central site may interfere with the sequential hopping process). Since $\hat{S}^+_i \hat{S}^-_i$ commutes with $\hat{S}^z_i$, the operators $\hat{h}_{i-1,1} \hat{h}_{i,1}$ and $\hat{h}_{i,2}$ effect exactly the same discrete derivative ($\sim \partial_x$) and therefore possess the same conserved quantities. As a result, the gates that we identify using the methods described in the main text are 'canonical' in the sense that they conserve all relevant $f(\mathbf{x})$ and all such diagonal operator insertions are absent; more complex gates can then of course be constructed by introducing such diagonal operators.

## Appendix E: Haar-random circuits and automaton dynamics

In the main text we confirmed that random circuits that conserve the finite list of multipole moments in Eq. (33) exhibit subsystem symmetry at long wavelengths. Here, we provide extra details pertaining to these simulations. In particular, we show how the Haar-random circuits can be simulated efficiently

by mapping to an effective automaton-like time evolution controlled by a 'transfer matrix'.

We work with spin-1/2 degrees of freedom that live on the sites of an $L \times L$ triangular lattice satisfying periodic boundary conditions. Acting on these degrees of freedom, we consider a quantum circuit with a random geometry composed of Haar random gates acting on clusters of sites. That is, rather than the standard "brickwork" geometry of gates, the location of each gate is chosen at random from a uniform distribution over the lattice (and one unit of time is defined as an extensive number of such random gate applications). For a given cluster acting on a region $\ell$, the unitary gate has the following structure

$$\hat{U}_\ell = \mathbb{1}_{\bar{\ell}} \otimes \bigoplus_\alpha u_\alpha \,, \qquad (E.1)$$

where $\bar{\ell}$ is the region of the lattice complementary to the gate region $\ell$, and $u_\alpha$ is an $n_\alpha \times n_\alpha$ random unitary matrix. The gate (E.1) is decomposed into blocks labeled by $\alpha$ according to their symmetry quantum numbers. That is, all states $|\alpha, m_\alpha\rangle$ (where $m_\alpha = 1, \ldots, n_\alpha$) are eigenstates of the multipole charges $\hat{\mathcal{Q}}[f]$ for $f$ belonging to the multipole group, $\hat{\mathcal{Q}}[f] |\alpha, m_\alpha\rangle = Q[f] |\alpha, m_\alpha\rangle$, and have the same eigenvalues $Q[f]$ for all $f$. Since all $\hat{\mathcal{Q}}[f]$ are diagonal in the $\hat{S}_i^z$ basis, we may take the decomposition (E.1) to be in the basis defined by product states of the form $\otimes_i |b_i\rangle$, with $|b_i\rangle \in \{|0\rangle, |1\rangle\}$ the eigenstates of $\hat{S}_i^z$ on site $i$, i.e., $\hat{S}_i^z |b\rangle = (-1)^b |b\rangle$.

We now show how Haar-averaged two-point correlation functions map onto stochastic automaton dynamics for operators that are diagonal in the $\hat{S}_i^z$ basis. The derivation is similar to that of Ref. [59], except that we work with states rather than vectorized operators, which makes the correspondence with automaton dynamics more crisp. Consider an infinite temperature two-point correlation function of the form

$$C_{ij}(t) = \overline{\text{Tr}\left[\hat{\rho}\hat{W}^\dagger(t)\hat{O}_i\hat{W}(t)\hat{O}_j\right]}, \qquad (E.2)$$

where the time evolution operator $\hat{W}(t)$ is given by a product of microscopic random unitaries (E.1), and $\hat{\rho} = \mathbb{1}/D$ is the infinite temperature density matrix ($D$ being the total dimension of the many-body Hilbert space). The overline denotes 'Haar averaging', i.e., averaging each block belonging to the gate (E.1) over the unitary group $\mathcal{U}(n_\alpha)$ with respect to the Haar measure. The corresponding ensemble of matrices is the circular unitary ensemble (CUE). If the operators are diagonal in the $\hat{S}_i^z$ basis, it is convenient to evaluate the trace in this basis:

$$C_{ij}(t) = \frac{1}{D} \sum_{\mathbf{s},\mathbf{s}'} O_i(\mathbf{s}') O_j(\mathbf{s}) \overline{\langle \mathbf{s}'| \hat{W}(t) |\mathbf{s}\rangle \langle \mathbf{s}| \hat{W}^\dagger(t) |\mathbf{s}'\rangle}, \qquad (E.3)$$

where $\mathbf{s}$ and $\mathbf{s}'$ represent eigenstates of $\hat{S}_i^z$. The quantity that is averaged over in Eq. (E.3) is interpreted as the probability that the system transitions from state $\mathbf{s}$ to $\mathbf{s}'$ after evolving the system for a time $t$:

$$P_{\mathbf{s}'\mathbf{s}}(t) \equiv \overline{\left|\langle \mathbf{s}'| \hat{W}(t) |\mathbf{s}\rangle\right|^2} \,. \qquad (E.4)$$

Suppose that each microscopic gate application is associated with a time $\tau$ (i.e., $L^2\tau = 1$ defines one unit of time). If a gate was applied on the region $\ell$ to evolve the system from time $t - \tau$ to $t$, then, inserting two resolutions of identity,

$$P_{\mathbf{s}'\mathbf{s}}(t) = \sum_{\mathbf{s}_1,\mathbf{s}_2} \overline{\langle \mathbf{s}'| \hat{U}_\ell |\mathbf{s}_1\rangle \langle \mathbf{s}_2| \hat{U}_\ell^\dagger |\mathbf{s}'\rangle} \times$$
$$\overline{\langle \mathbf{s}_1| \hat{W}(t-\tau) |\mathbf{s}\rangle \langle \mathbf{s}| \hat{W}^\dagger(t-\tau) |\mathbf{s}_2\rangle} \,. \quad (E.5)$$

Note that, since each gate is drawn from an independent distribution, the CUE average has decomposed into two separate averages. Since the gate (E.1) acts as the identity outside of $\ell$, we immediately observe that the matrix elements enforce that the states $\mathbf{s}_1$ and $\mathbf{s}_2$ must coincide in $\bar{\ell}$. We now perform the average on the top line exactly using the one-fold Haar channel: for each block, $\Phi[A] \equiv \overline{\hat{U}\hat{A}\hat{U}^\dagger} = d^{-1}\text{Tr}[\hat{A}]\mathbb{1}$ [84] (with $d$ the dimension of the random matrix $\hat{U}$)

$$\overline{\langle \mathbf{s}'| \hat{U}_\ell |\mathbf{s}_1\rangle \langle \mathbf{s}_2| \hat{U}_\ell^\dagger |\mathbf{s}'\rangle} =$$
$$\langle \mathbf{s}_1|\mathbf{s}_2\rangle_{\bar{\ell}} \sum_\alpha \frac{1}{n_\alpha} \langle \mathbf{s}_2| \hat{P}^\alpha |\mathbf{s}_1\rangle_\ell \langle \mathbf{s}'| \hat{P}^\alpha |\mathbf{s}'\rangle_\ell \,, \quad (E.6)$$

where the subscripts $\ell$ and $\bar{\ell}$ denote the region of the lattice on which the inner products are evaluated, and $n_\alpha$ is the number of states in the symmetry block $\alpha$. Since the projectors can also be chosen to be diagonal in the $\hat{S}_i^z$ basis, each term under the summation vanishes if $|\mathbf{s}_1\rangle_\ell$ or $|\mathbf{s}_2\rangle_\ell$ does not belonging to the block $\alpha$. If both belong to the block $\alpha$, then the two states must coincide. We can therefore eliminate $\mathbf{s}_2$ from (E.5) and simplify (E.6) to give

$$P_{\mathbf{s}'\mathbf{s}}(t) = \sum_{\mathbf{s}_1} T_{\mathbf{s}'\mathbf{s}_1}^\ell P_{\mathbf{s}_1\mathbf{s}}(t - \tau) \qquad (E.7)$$

where we defined the 'transfer matrix'

$$T_{\mathbf{s}'\mathbf{s}}^\ell = \sum_\alpha \frac{1}{n_\alpha} \langle \mathbf{s}| \hat{P}^\alpha |\mathbf{s}\rangle_\ell \langle \mathbf{s}'| \hat{P}^\alpha |\mathbf{s}'\rangle_\ell \,. \qquad (E.8)$$

The evolution from $\mathbf{s} \to \mathbf{s}'$ can therefore be decomposed into a sequence of such transfer matrices, each of which incorporates the effect of a gate application. The transfer matrix (E.8) is the state version of the operator transfer matrix derived in Ref. [59]; it checks which block the input state belongs to and sends it to a mixture of all other local states belonging to the same block with uniform probability determined by the size of the block: $1/n_\alpha$. The correlation function (E.2) can then be evaluated efficiently by performing a stochastic automaton time evolution, where $\hat{S}_i^z$ eigenstates $|\mathbf{s}\rangle$ are sent to other eigenstates $|\mathbf{s}'\rangle$ according to the transition probabilities determined by the transfer matrix (E.8). To make sure that the gate (36) (which microscopically breaks the subsystem symmetry) is included in the circuit, we work with clusters of sites of radius $\ell$ satisfying $2\ell = \sqrt{19} + \epsilon$, centered on bonds of the lattice.

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
