# Peer review of "Multipole groups and fracton phenomena on arbitrary crystalline lattices"

_SciPost Physics_

## Round 1 · Referee Report · Anonymous (Referee 1) · 2023-7-8

Report

This is a clear and mathematically sound paper that provides a framework for determining sets of multipole symmetries that are consistent with any given crystal lattice. In particular, the paper answers the question "given a crystal lattice and a specific conservation of a multipole moment, which other multipole moments must be conserved?" (akin to the well-know fact that the dipole moment conservation implies charge conservation in translationally invariant systems). The authors' framework is generic and can be applied to both Bravais lattices and lattices with more than one site in the unit cell. The authors apply their framework to several 2D lattices, explicitly summarizing the results. It is noteworthy that, in addition to explaining certain known results within their framework, the authors have discovered two new peculiar phenomena: 1) a finite number of global multipole symmetries can induce emergent subsystem symmetries (if the interaction range is small enough); 2) a translationally invariant system can conserve certain multipole moments without conserving the monopole charge (but instead conserves a vector charge).

In my opinion, the paper satisfies all the required criteria, and I recommend it for publication in SciPost Physics.

I would be happy if the authors could clarify several points for me, though: 1) A lattice only allows for finite translations. Why is it then appropriate to consider infinitesimal translations (with the procedure from Appendix C) to get the lower-degree irreps? 2) If an irrep of the extended point group is more-than-1-dimensional, what does it mean physically in terms of the conservation laws of the model? 3) In Fig. 3, what happens if we impose conservation of, say, $x^4+y^4$ and $x^3y+y^3x$, i.e., the ones pointing to different gray clouds. Which cubic terms would be conserved then? This remained unclear to me both from the text and from the figure. 4) Is it possible that conservation of a certain multipole moment on some lattice is completely forbidden? Or one can always enforce conservation of any given multipole moment on any given lattice (and then find all the descendant multipole moments that must be conserved)? 5) Is it possible that conservation of certain combinations of lower-degree multipole moments on some lattice would imply conservation of a higher-degree moment?

I also have two minor cosmetic suggestions (but I leave the decision up to the authors): a) In Fig. 1, it would be more informative to also specify the angles between the lattice vectors in the picture (i.e., $\gamma= \pi/2$, $\gamma \neq \pi/2$, or $\gamma = \pi/3$). b) In Eq. (8), $C$ denotes both a rotation and a basis site. Perhaps one could slightly change the notation by, e.g., changing the font of one of the $C$'s.

  • validity: high
  • significance: high
  • originality: high
  • clarity: high
  • formatting: perfect
  • grammar: perfect

Author:  Oliver Hart  on 2023-08-03  [id 3869]

(in reply to Report 1 on 2023-07-08)

This is a clear and mathematically sound paper that provides a framework for determining sets of multipole symmetries that are consistent with any given crystal lattice. In particular, the paper answers the question "given a crystal lattice and a specific conservation of a multipole moment, which other multipole moments must be conserved?" (akin to the well-know fact that the dipole moment conservation implies charge conservation in translationally invariant systems). The authors' framework is generic and can be applied to both Bravais lattices and lattices with more than one site in the unit cell. The authors apply their framework to several 2D lattices, explicitly summarizing the results. It is noteworthy that, in addition to explaining certain known results within their framework, the authors have discovered two new peculiar phenomena: 1) a finite number of global multipole symmetries can induce emergent subsystem symmetries (if the interaction range is small enough); 2) a translationally invariant system can conserve certain multipole moments without conserving the monopole charge (but instead conserves a vector charge).

In my opinion, the paper satisfies all the required criteria, and I recommend it for publication in SciPost Physics.

Reply: We thank the referee for their summary and positive evaluation.

1) A lattice only allows for finite translations. Why is it then appropriate to consider infinitesimal translations (with the procedure from Appendix C) to get the lower-degree irreps?

Reply: Appendix C is not a procedure for considering infinitesimal translations; it is a proof that infinitesimal translations and finite translations generate the same lower-degree irreps. The proof is somewhat technical, but the outline is roughly the following. For a polynomial of degree $n$, translating by each (discrete) lattice vector up to $n+1$ times produces lower-degree polynomials which are distinct and, as it turns out, linearly independent. Similarly, taking infinitesimal translations along each lattice vector directions produces some set of linearly independent lower-degree polynomials at each degree. Certainly repeated infinitesimal translations will eventually produce finite translations, so the polynomials produced via infinitesimal translations include those produced by finite translations. Showing the opposite inclusion is the main result of Appendix C.

2) If an irrep of the extended point group is more-than-1-dimensional, what does it mean physically in terms of the conservation laws of the model?

Reply: This means that in the presence of the space group symmetry, certain multipole moments have multiple components that are either all conserved, or none are (as long as the space group symmetry is present). For an intuitive example, consider dipole moment conservation on the square lattice. The fact that $x$ belongs to the 2D irrep $\lbrace x, y\rbrace$ of $\mathsf{D}_4$ means that if the $x$ component of dipole moment is conserved, point group symmetry requires the $y$ component to be conserved as well.

3) In Fig. 3, what happens if we impose conservation of, say, $x^4+y^4$ and $x^3y+y^3x$, i.e., the ones pointing to different gray clouds. Which cubic terms would be conserved then? This remained unclear to me both from the text and from the figure.

Reply: The span of $\lbrace x^3, y^3\rbrace$ and $\lbrace x^3 + 3xy^2, y^3+3x^2y\rbrace$ would also be conserved. The clouds do not change this interpretation; they just indicate that the list of cubic terms in the figure is redundant, i.e., the left-hand cloud spans all possible cubic terms, but so does the right-hand cloud.

The reason for the clouds is that if we choose a fixed basis for the cubic polynomials irreps, some of the quartic polynomials will conserve not one, not both, but a specific nontrivial linear combination of the (degenerate) cubic polynomial irreps. The clouds were the best visual representation we had available to show that fact. We have added a short description of this to Sec. II.B.3.

4) Is it possible that conservation of a certain multipole moment on some lattice is completely forbidden? Or one can always enforce conservation of any given multipole moment on any given lattice (and then find all the descendant multipole moments that must be conserved)?

Reply: It is not possible to completely forbid the conservation of a multipole moment via lattice symmetries. Given any fixed multipole moment, one can just find its orbit under the point group and then find the translation descendants.

5) Is it possible that conservation of certain combinations of lower-degree multipole moments on some lattice would imply conservation of a higher-degree moment?

Reply: We are unaware of any reason for higher-degree moments to be exactly conserved; certainly lattice symmetries cannot be responsible for this. However, as our examples show, lower-degree moments can lead to the emergent conservation of higher-degree moments, i.e., at low enough energies higher-degree moments will be approximately conserved. This is how, e.g., we obtain emergent subsystem symmetries from a fixed list of low-degree moments.

I also have two minor cosmetic suggestions (but I leave the decision up to the authors): a) In Fig. 1, it would be more informative to also specify the angles between the lattice vectors in the picture (i.e., $\gamma=\pi/2$, $\gamma\neq\pi/2$, or $\gamma=\pi/3$). b) In Eq. (8), C denotes both a rotation and a basis site. Perhaps one could slightly change the notation by, e.g., changing the font of one of the C's.

Reply: We thank the referee for their useful suggestions. We have updated Fig. 1 and its caption accordingly, and added a comment on overloading the letter $C$.

---

## Round 1 · Referee Report · Anonymous (Referee 2) · 2023-7-11

Strengths

1 Clearly formulated mathematical framework to explore multipole symmetries on arbitrary lattices.

2 Novel findings based on the provided framework: (i) finite number of global multipole symmetries can induce emergent subsystem symmetries and (ii) translationally invariant system can conserve certain multipole moments without conserving the monopole charge.

Report

In this work, the authors provide a framework to explore multipole symmetries. While previous studies have mostly focused on continuum systems and hyper cubic lattices, the present framework can be applied to on arbitrary crystal lattices. The manuscript is clearly written and the results are sound. Interestingly, an exact multipolar symmetry on the breathing kagome lattice is found that does not include conservation of charge but instead conserves a vector charge.

I recommend publication in SciPost Physics.

Minor "historical" comment: Refs. [32][19][40] all appeared within a couple of days on the arXiv and independently derived subdhiffusive transport for dipole conserving systems.
  • validity: top
  • significance: high
  • originality: high
  • clarity: top
  • formatting: excellent
  • grammar: perfect

Author:  Oliver Hart  on 2023-08-03  [id 3868]

(in reply to Report 2 on 2023-07-11)

In this work, the authors provide a framework to explore multipole symmetries. While previous studies have mostly focused on continuum systems and hyper cubic lattices, the present framework can be applied to on arbitrary crystal lattices. The manuscript is clearly written and the results are sound. Interestingly, an exact multipolar symmetry on the breathing kagome lattice is found that does not include conservation of charge but instead conserves a vector charge.

I recommend publication in SciPost Physics.

Reply: We thank the referee for their summary and positive evaluation.

Minor "historical" comment: Refs. [32][19][40] all appeared within a couple of days on the arXiv and independently derived subdhiffusive transport for dipole conserving systems.

Reply: The references in question have now become numbers [35], [20] and [36], respectively, in the new version. Reference numbers in what follows refer to the new version with these updated numbers.

We thank the referee for this comment. In regards to the chronology of subdiffusion in fractonic systems, we note that Ref. [34] was posted to the arXiv on 24 July 2019, Ref. [35] on 20 March 2020, Ref. [20] on 31 March, 2020, and Ref. [36] on 1 April 2020. Certainly all papers are important, and we have ensured that all are cited prominently in the context of fracton hydrodynamics.

---

## Editorial Decision

resubmitted